# Low potassium activation of proximal mTOR/AKT signaling is mediated by Kir4.2

Yahua Zhang[1,2], Fabian Bock[1,2], Mohammed Ferdaus[3], Juan Pablo Arroyo [1,2], Kristie L Rose[4,5], Purvi Patel[5], Jerod S. Denton[3], Eric Delpire[3], Alan M. Weinstein[6], Ming-Zhi Zhang [1,2], Raymond C. Harris[1,2,7] & Andrew S. Terker [1,2] ✉

The renal epithelium is sensitive to changes in blood potassium ($K^+$). We identify the basolateral $K^+$ channel, Kir4.2, as a mediator of the proximal tubule response to $K^+$ deficiency. Mice lacking Kir4.2 have a compensated baseline phenotype whereby they increase their distal transport burden to maintain homeostasis. Upon dietary $K^+$ depletion, knockout animals decompensate as evidenced by increased urinary $K^+$ excretion and development of a proximal renal tubular acidosis. Potassium wasting is not proximal in origin but is caused by higher ENaC activity and depends upon increased distal sodium delivery. Three-dimensional imaging reveals Kir4.2 knockouts fail to undergo proximal tubule expansion, while the distal convoluted tubule response is exaggerated. AKT signaling mediates the dietary $K^+$ response, which is blunted in Kir4.2 knockouts. Lastly, we demonstrate in isolated tubules that AKT phosphorylation in response to low $K^+$ depends upon mTORC2 activation by secondary changes in $Cl^-$ transport. Data support a proximal role for cell $Cl^-$ which, as it does along the distal nephron, responds to $K^+$ changes to activate kinase signaling.

The kidney is exquisitely sensitive to alterations in systemic potassium ($K^+$) balance. Changes in $K^+$ homeostasis, whether from dietary, pharmacological, or genetic causes, have potent effects on kidney epithelial cell function. For decades it has been known that reductions in systemic $K^+$ levels induce profound kidney enlargement[1]. Renal hypertrophy is concomitant with metabolic changes, including increased gluconeogenesis and ammoniagenesis[2,3], and alterations in ion transport. This not only includes adjustments in transport to reduce $K^+$ losses but also strong activation of $Na^+$ reabsorptive pathways along both proximal and distal segments[4].

Work from the last several years has identified detailed mechanisms governing the distal activation of $Na^+$ transport in response to low $K^+$ states, which is important in the generation of salt-sensitive hypertension[5–8]. The ability of the distal convoluted tubule (DCT) to increase its $Na^+$ reabsorptive load depends on the activity of the inwardly rectifying $K^+$ channel, Kir4.1, which is present along the basolateral membrane of DCT cells and forms functional heterotetramers with Kir5.1. Changes in Kir4.1/5.1 flux affect membrane potential, which ultimately activates the With no lysine (WNK) kinase/ Ste20p-related proline alanine-rich kinase (SPAK) cascade, stimulating the sodium chloride cotransporter (NCC) to increase apical $Na^+$ reabsorption[6]. Loss of Kir4.1 in mice abrogates this response and causes disease in humans[5,9].

Much less attention has been focused on proximal mechanisms in recent years. The proximal tubule (PT) epithelium comprises the bulk of renal mass and is the major site of ammoniagenesis, gluconeogenesis, and $Na^+$ and $K^+$ reabsorption in the kidney. Molecular signaling underlying the $K^+$-sensitivity of these proximal pathways remains

[1]Division of Nephrology, Department of Medicine, Vanderbilt University Medical Center, Nashville, TN, USA. [2]Vanderbilt Center for Kidney Disease, Nashville, TN, USA. [3]Department of Anesthesiology, Vanderbilt University Medical Center, Nashville, TN, USA. [4]Department of Biochemistry, Vanderbilt University School of Medicine, Nashville, TN, USA. [5]Mass Spectrometry Research Center, Vanderbilt University School of Medicine, Nashville, TN, USA. [6]Department of Physiology and Biophysics, Weil Medical College, New York, NY, USA. [7]Department of Veterans Affairs, Tennessee Valley Healthcare System, Nashville, TN, USA. ✉e-mail: Andrew.s.terker@vumc.org

largely unexplored. While Kir4.1/5.1 mediates the distal K⁺ response, the identity of the K⁺ channel mediating proximal responses has not been reported. Bignon et al. described an essential role for the Kir4.1-related channel, Kir4.2, in modulating the PT response to acid-base derangements[10]. Kir4.2 is expressed along the PT basolateral membrane, where it maintains the resting membrane potential. They demonstrated Kir4.2 knockout (Kir4.2$^{-/-}$) animals develop a proximal renal tubular acidosis (RTA) when challenged with an acid load due to a failure to appropriately induce ammoniagenesis and increase net acid excretion. We previously reported that deletion of Kir4.2 prevents kidney injury caused by reductions in systemic K⁺ levels, but whether or not this channel mediates the physiological response to low K⁺ is unknown.

Also poorly defined are the downstream K⁺-sensitive signaling pathways affecting proximal cell growth and transport. The mechanistic/mammalian target of rapamycin (mTOR) is a master regulator of cell growth, differentiation, and metabolism[11,12]. As a kinase, it exists at the core of a signaling complex existing in two distinct forms, mTOR complex (mTORC) 1 and mTORC2. Phosphorylation of one of the most well-characterized mTORC2 targets, AKT, occurs in response to growth factor stimulation and altered nutrient availability to drive changes in cell proliferation, cytoskeletal rearrangements, and energy metabolism[13]. While mTOR signaling has been shown to be involved in distal nephron electrolyte balance, these effects are independent of AKT[14–16]. While a few studies have focused on AKT as a regulator of ion transport[17,18], our knowledge of AKT function in the kidney is surprisingly minimal. Its role in the proximal low K⁺ response has not been studied.

Here we tested the hypothesis that Kir4.2 serves as a mediator of reduced basolateral K⁺ and PT epithelial function and investigated the signaling pathways involved. We observed that the deletion of Kir4.2 in mice reduces proximal Na⁺ transport, an effect that is compensated by increased distal transport pathways. Upon dietary K⁺ depletion, Kir4.2$^{-/-}$ animals experience urinary K⁺ wasting that occurs distally via the epithelial Na⁺ channel (ENaC). Ex vivo isolated tubule culture identifies mTORC2/AKT signaling downstream of Kir4.2 as essential for the proximal response to low K⁺. The system is activated by secondary increases in basolateral (Cl⁻) transport in response to low K⁺ exposure. mTORC2/AKT target phosphorylation is reduced following Kir4.2 deletion and low K⁺-induced kidney enlargement is absent in Kir4.2$^{-/-}$ mice. The findings identify an integral role for Kir4.2 in the PT K⁺ response. They also support a role for increased PT basolateral Cl⁻ transport in the activation of mTORC2/AKT to coordinate cell growth and electrolyte transport.

## Results

### Kir4.2 deletion causes K⁺ wasting following dietary K⁺ depletion

A previous report of independently generated Kir4.2$^{-/-}$ animals displayed minimal electrolyte abnormalities at baseline[10]. To determine if this was the case in our Kir4.2$^{-/-}$ mice, we measured baseline blood and urine electrolytes in control (Kir4.2$^{+/+}$) and Kir4.2 knockouts. Consistent with this previous report, our knockout animals did not display differences in blood or urine electrolytes except for slightly higher blood Cl⁻ levels compared to controls (Supplementary Fig 1a–d). Differences in urinary excretion of individual electrolytes and baseline blood pressure measurements were also not statistically significant (Supplementary Fig 1e–i).

As the PT reabsorbs the majority of filtered K⁺, we next investigated how deletion of Kir4.2 affected kidney electrolyte handling under low K⁺ stress. After consuming a K⁺-deficient (0 K) diet for four days, blood electrolyte analysis revealed knockouts had lower blood K⁺ levels compared to controls (Fig. 1a). They exhibited higher blood Na⁺ and Cl⁻ levels along with reduced blood HCO₃⁻, findings consistent with a proximal renal tubular acidosis (RTA) (Fig. 1b–d). Aside from modestly elevated blood Cl⁻ in the knockout group, large differences in blood electrolytes were not detected between genotypes following

consumption of a NaCl-deficient diet for four days suggesting the channel is essential for the renal response to dietary K⁺, but not Na⁺, restriction (Supplementary Fig 2a–d).

These findings were suggestive of renal K⁺ wasting in our knockout animals, so we performed metabolic cage experiments to measure overnight urine electrolyte excretion during the animals' active period. While animals from both genotypes reduced their urine K⁺ concentrations and total daily urine K⁺ excretion on the 0 K diet, both parameters remained higher in Kir4.2$^{-/-}$ mice compared to controls, although differences did not achieve significance by post-hoc testing (Fig. 1e, f). Cumulative urine K⁺ losses were significantly greater in knockout animals supporting a K⁺ wasting phenomenon underlying their reduced blood K⁺ (Fig. 1g). Urinary K⁺ wasting could also be detected in Kir4.2$^{-/-}$ mice via daily spot urine collections validating spot urine use for future studies (Fig. 1h). Daily urine Na⁺ concentrations and total daily urine Na⁺ excretion were not different between genotypes at individual timepoints by post-hoc testing either, but there was a trend towards higher total urine Na⁺ excretion in knockouts on the first day of 0 K treatment (Fig. 1i, j). Pooling normalized data revealed higher urine Na⁺ in Kir4.2$^{-/-}$ mice after one day of 0 K consumption (Fig. 1k). Knockouts also excreted more free water during this first day of treatment (Fig. 1l).

### Potassium wasting in Kir4.2 knockouts is a distal phenomenon mediated by ENaC

While the majority of the filtered electrolyte load, including K⁺, is reabsorbed proximally, previous reports have suggested K⁺ wasting in proximal RTA occurs distally[19]. K⁺ secretion along the connecting tubule (CNT) is mediated by apical K⁺ channels, which secrete K⁺ in response to a negative lumen potential generated by Na⁺ reabsorption via the epithelial Na⁺ channel (ENaC). To determine if K⁺ wasting observed in knockout animals is dependent on a Na⁺ load, we again treated animals of both genotypes with a K⁺-deficient diet but varied dietary Na⁺ in the setting of 0 K. Removal of dietary Na⁺ from the 0 K diet reduced K⁺ wasting in Kir4.2$^{-/-}$ mice on a 0 K diet compared to knockouts consuming a 0.3% Na⁺/0 K diet (Fig. 2a, b).

To formally test if Na⁺ reabsorption via ENaC underlies excessive distal K⁺ secretion, we treated knockout mice on a 0 K diet with either regular drinking water or water supplemented with the ENaC inhibitor, amiloride (75 mg/L). As presented in Fig. 2c, amiloride treatment prevented K⁺ wasting observed in knockout animals on normal drinking water, but it did not cause a significant change in urine Na⁺ excretion (Supplementary Fig 3a). Amiloride did not have a detectable effect on Kir4.2$^{+/+}$ animals on 0 K while it returned Kir4.2$^{-/-}$ urine K⁺ to levels that were indistinguishable from Kir4.2$^{+/+}$ mice suggesting it completely prevented K⁺ wasting (Fig. 2d). Correcting the K⁺ wasting also ameliorated the hypokalemia caused by the 0 K diet (Fig. 2e). Blood Na⁺ and blood Cl⁻ were reduced as well suggesting amiloride prevented the water wasting observed in knockouts (Fig. 2f, g). This was supported by higher urine osmolality in the amiloride-treated group (Supplementary Fig 3b). Amiloride did not impact blood HCO₃⁻ levels suggesting acid-base disturbances were primarily of proximal origin (Fig. 2h). A reduction in Kir4.2$^{-/-}$ urine K⁺ was also observed following treatment with the chemically distinct ENaC inhibitor, triamterene, to further demonstrate the role of ENaC in mediating the K⁺ wasting observed in knockout animals (Supplementary Fig 3c).

### Stimulated distal transport pathways compensate for proximal Na⁺ losses in Kir4.2$^{-/-}$ mice

These data supported increased ENaC activity in knockout animals. Therefore, we next measured abundance of ENaC subunits. While we could not detect differences in αENaC, we did observe increased cleavage of the γ subunit (Fig. 3a, d, e). This was accompanied by elevated plasma aldosterone levels in knockouts (Fig. 3c). NCC is also essential for distal nephron regulation of K⁺ balance[7,20]. We detected

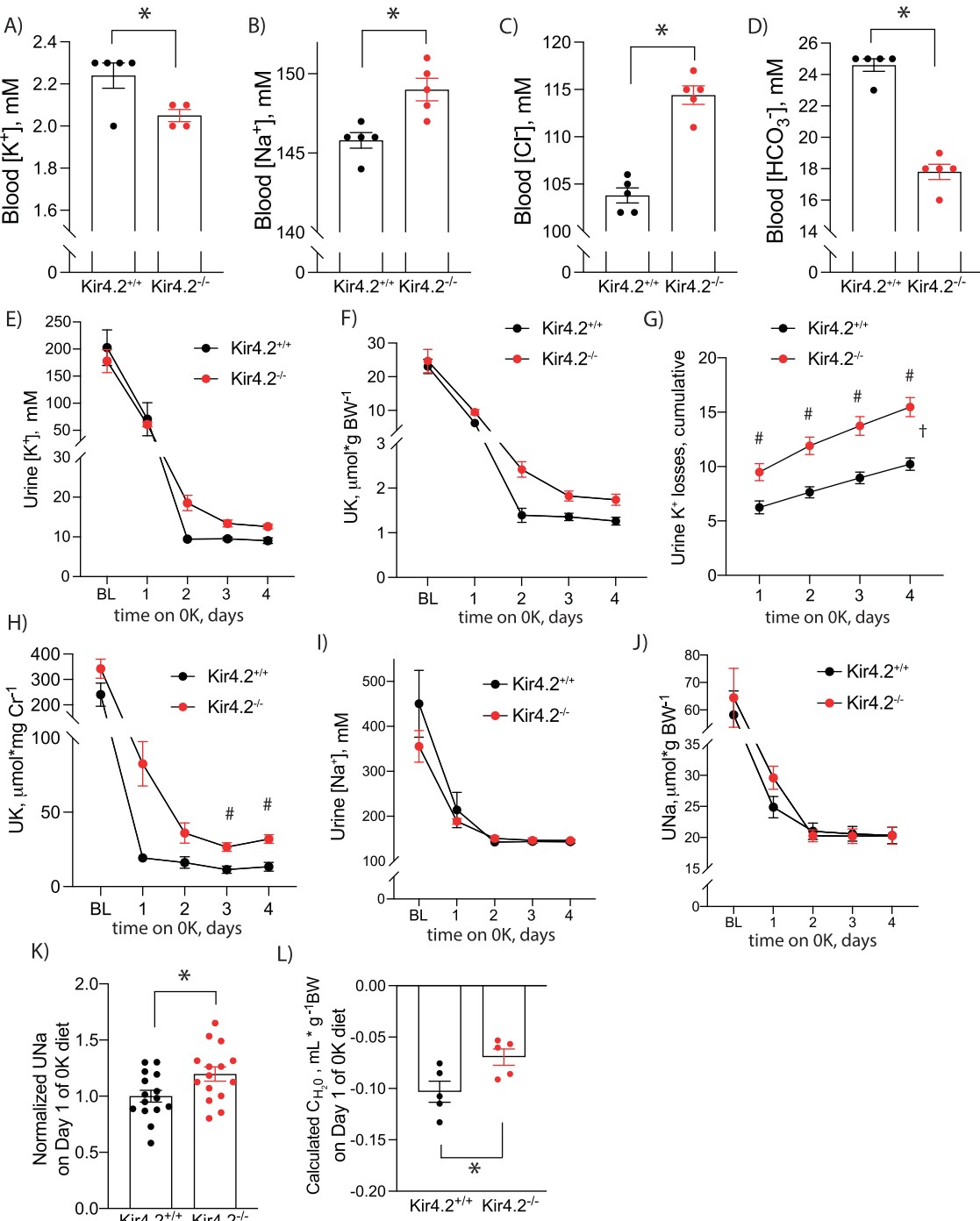

**Fig. 1 | Blood electrolytes in Kir4.2$^{+/+}$ and Kir4.2$^{-/-}$ animals following four days on 0 K diet.** Blood (**A**) K$^+$ ($p = 0.034$), (**B**) Na$^+$ ($p = 0.0059$), (**C**) Cl$^-$ ($p = 3 \times 10^{-5}$), and (**D**) HCO$_3^-$ ($1.5 \times 10^{-4}$) values in Kir4.2$^{+/+}$ and Kir4.2$^{-/-}$ animals after four days on 0 K diet. Daily urine K$^+$ (**E**) concentration and (**F**) excretion at baseline and on 0 K diet for indicated time. **G** Cumulative urine K$^+$ losses on 0 K diet ($p = 0.0068, 0.0002, <0.0001, <0.0001$). **H** Urine K$^+$ excretion normalized to creatinine from spot urines collected from animals at baseline and on 0 K diet for indicated times ($p = 0.018, 0.0096$). Daily urine Na$^+$ (**I**) concentration and (**J**) excretion at baseline and on 0 K diet for indicated time. **K** Normalized daily hour urine Na$^+$ excretion in mice consuming a 0 K diet for 1 day ($p = 0.024$). **L** Calculated free water clearance for mice consuming a 0 K diet for 1 day ($p = 0.032$). $N = 5$ per group for (**A**−**D**) and (**L**) except (**A**) where $N = 5$ for Kir4.2$^{+/+}$ and 4 for Kir4.2$^{-/-}$. One knockout blood K$^+$ sample was excluded for the presence of gross hemolysis. $N = 10$ per group for (**E**−**J**) and 15 per group for (**K**). *$P < 0.05$ by unpaired $t$ test. $^\#P < 0.05$ for genotype difference at indicated timepoints by two-way ANOVA with repeated measures followed by Sidak's multiple comparison test. $^\dagger P < 0.05$ for interaction between treatment and time variables by two-way ANOVA with repeated measures. All tests were two-sided. Data presented as mean ± sem.

increased phosphorylation (pNCC) and total abundance of NCC in knockout mice (Fig. 3b, f, g). Total NKCC2 abundance was also increased in Kir4.2$^{-/-}$ animals compared with controls (Fig. 3a, h). Abundance of the distal nephron basolateral K channel, Kir4.1, did not differ between genotypes (Supplementary Fig 4). To confirm that

increased pNCC abundance reflected increased function in vivo, we performed an HCTZ-response test in Kir4.2$^{+/+}$ and Kir4.2$^{-/-}$ animals. Knockouts demonstrated increased natriuretic and kaliuretic responses to acute HCTZ treatment (Fig. 3i, j). Similarly, knockouts had an increased kaliuretic response to acute furosemide treatment, though

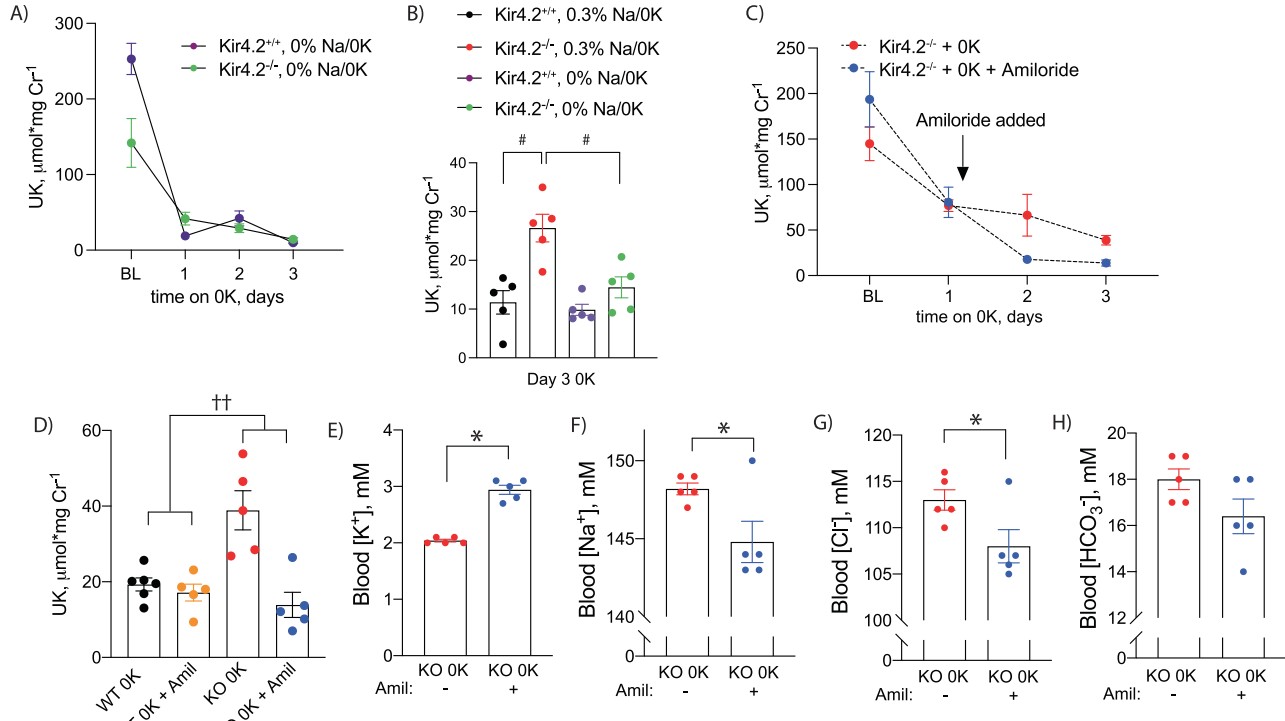

**Fig. 2 | Effects of dietary Na$^+$ intake and ENaC inhibition on urine K$^+$ excretion in Kir4.2$^{+/+}$ and Kir4.2$^{-/-}$ animals. A** Urine K$^+$ excretion at baseline and on 0 K diet for indicated timepoints in Kir4.2$^{+/+}$ and Kir4.2$^{-/-}$ mice. Diet contained 0% Na$^+$ with 0 K (purple and green). **B** Urine K$^+$ excretion on day three of 0 K from animals treated with normal Na$^+$ or as in (**A**). Note black and red data (both genotypes on 0.3% Na$^+$) are from panel 1 h ($p = 0.0011$ and $0.0082$). Urine (**C**) K$^+$ excretion at baseline and on 0 K diet in Kir4.2$^{-/-}$ mice. Amiloride was added to drinking water at indicated time point ($p = 0.02$). **D** Comparison of urine K$^+$ excretion between Kir4.2$^{+/+}$ and Kir4.2$^{-/-}$ mice after three days of consuming a 0 K diet with and without amiloride treatment

($p = 0.0027$ for interaction). Blood (**E**) K$^+$ ($p = 5 \times 10^{-6}$) (**F**) Na$^+$ ($p = 0.038$), (**G**) Cl$^-$ ($p = 0.044$), and (**H**) HCO$_3^-$ from Kir4.2$^{-/-}$ mice treated as in (**C**). $N = 5$ for all, except day 1 of 0 K in 2a where $n = 2$ and 4 (3 Kir4.2$^{+/+}$ and 1 Kir4.2$^{-/-}$ animals did not consume diet and so data were not included in analysis), and $N = 6$ for WT 0 K in (**D**). #, $P < 0.05$ for genotype difference at indicated timepoints by two-way ANOVA with or without repeated measures followed by Sidak's multiple comparison test. *$P < 0.05$ by unpaired $t$ test. $^{††}p < 0.05$ for interaction by two-way ANOVA. All tests were two-sided. Data presented as mean ± sem.

---

the increase in natriuretic response did not quite meet the threshold for statistical significance (Fig. 3k, l). We measured the calciuretic response to furosemide, as loop diuretics also increase urine calcium excretion, and observed knockouts had an increased calcium response further supporting their increased NKCC2 activity (Fig. 3m). Overall, functional data corroborate Western blot results and support increased ENaC, NCC, and NKCC2 activity in knockout mice.

To determine if distally increased Na$^+$ reabsorption offsets proximal reductions, we performed Li$^+$ clearance (Cl$_{Li}$) studies. These studies revealed that differences in renal Na$^+$ clearance (Cl$_{Na}$) or Cl$_{Li}$ by themselves did not differ between genotypes, but the Cl$_{Li}$ to Cl$_{Na}$ ratio was increased in Kir4.2$^{-/-}$ mice consistent with reduced proximal Na$^+$ transport (Fig. 3n–p).

### Low K$^+$ stress fails to induce PT expansion while the distal response is exaggerated in Kir4.2 knockout animals

Low K$^+$ conditions impose significant metabolic stress along the nephron. This includes increased Na$^+$ reabsorption along both proximal and distal nephron segments[4] and activation of gluconeogenic and ammoniagenic pathways[2,3]. Coupled with this increased metabolic activity, the hypokalemic kidney undergoes a significant growth response[1]. We observed increased kidney weight in control animals following four days of 0 K treatment; mass further increased after eight days of K$^+$ deficiency (Fig. 4a). A detectable increase in gross kidney mass was absent in mice lacking Kir4.2. Nephron segment expansion in response to low K$^+$ has been well-documented along both proximal and distal nephron segments[21]. Despite absent whole kidney enlargement, optical clearing followed by three-dimensional confocal imaging

revealed increased DCT volume in Kir4.2 knockouts following 0 K feeding compared to controls (Fig. 4b, c). We next measured PT diameter and cortical thickness using a similar approach based on LTL staining. These analyses revealed knockout animals on 0 K had significantly reduced cortical thickness and PT diameter compared to Kir4.2$^{+/+}$ mice (Fig. 4d–g, Supplementary Fig 5).

### AKT controls renal Na$^+$ and K$^+$ handling and its regulation is disrupted following Kir4.2 deletion

Defective proximal expansion in Kir4.2$^{-/-}$ animals, prompted us to examine activation of growth signaling pathways in our mice. Notably, we detected increased phosphorylation of AKT (pAKT) at serine 473 in control animals on 0 K, a finding that was not observed in knockouts (Fig. 5a, Supplementary Fig 6a, b). AKT isoform-specific phospho-antibodies indicated both pAKT1 and pAKT2 were increased in response to low K$^+$ (Fig. 5b, Supplementary Fig 6c, d). Consistent with previous reports[22], AKT3 was not detected in the kidney (Supplementary Fig 6e). In Kir4.2$^{+/+}$ animals, AKT staining adopted a punctate pattern in the PT both at baseline and following 0 K feeding (Fig. 5c, d, Supplementary Fig 6f, g, Supplementary Fig 7). AKT is generally known to be a key regulator of cell growth and metabolism, but its role in the kidney and if it is involved in ion transport regulation, is less well-defined. To determine this, we measured the renal response to treatment with the AKT inhibitor, MK2206. Wild-type animals exhibited a natriuretic and kaliuretic response in the four hours following acute MK2206 administration at both 50 mg/kg and 25 mg/kg (Fig. 5e, Supplementary Fig 8a). This response was exaggerated following the consumption of a 0 K diet (Fig. 5f, Supplementary Fig 8b, c). While a

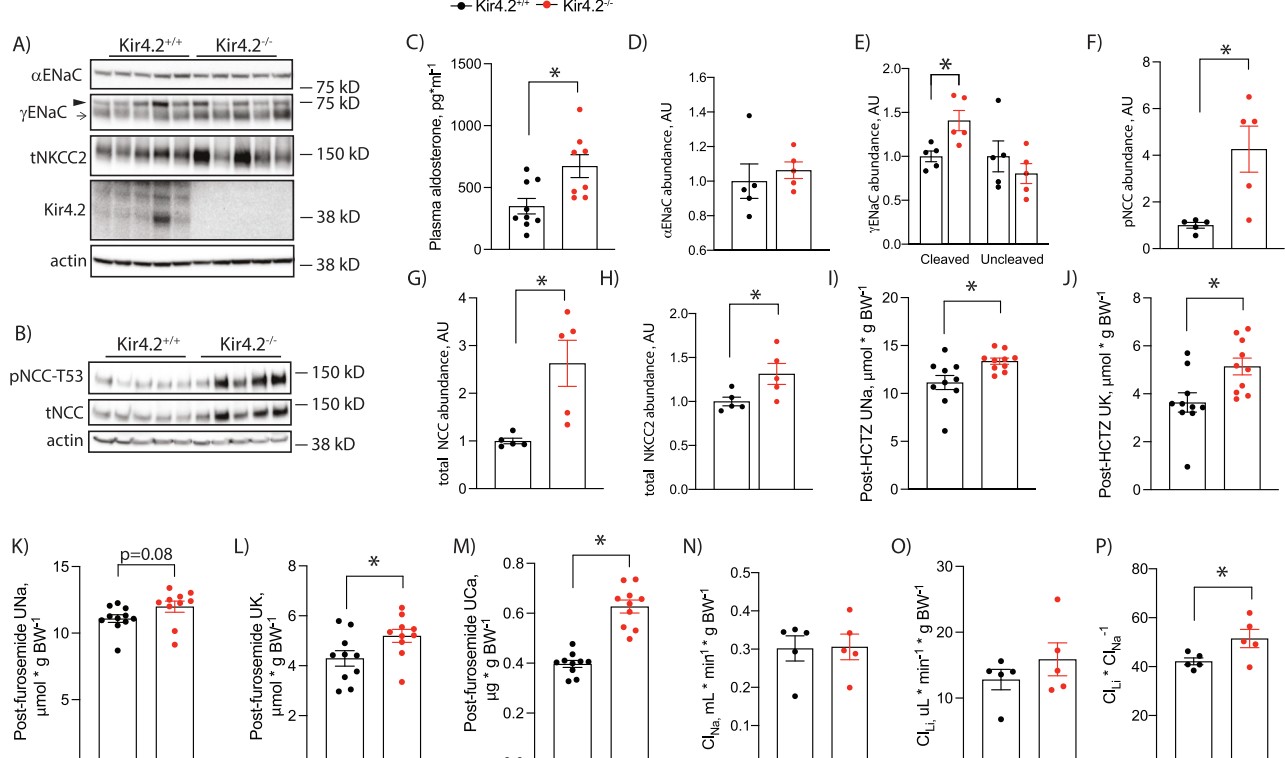

**Fig. 3 | Effects of Kir4.2 deletion on distal renal epithelial transporters/channels in mice on a normal diet.** Total kidney abundances of (**A**) αENaC, γENaC, total NKCC2, Kir4.2 and (**B**) pNCC-T53 and total NCC in Kir4.2$^{+/+}$ and Kir4.2$^{-/-}$ animals. ► uncleaved γENaC, → cleaved γENaC. **C** Plasma aldosterone concentrations in Kir4.2$^{+/+}$ and Kir4.2$^{-/-}$ animals ($p = 0.0096$). **D–H** Quantification for blots shown in (**A**) and (**B**) ($p = 0.013$ for (**E**), 0.011 for (**F**), 0.010 for (**G**), and 0.041 for (**H**)). **I** Natriuretic and (**J**) kaliuretic responses of Kir4.2$^{+/+}$ and Kir4.2$^{-/-}$ mice following acute HCTZ treatment ($p = 0.013$ for I and 0.012 for (**J**). **K** Natriuretic, (**L**) kaliuretic, and (**M**) calciuretic responses of Kir4.2$^{+/+}$ and Kir4.2$^{-/-}$ mice following acute furosemide treatment ($p = 0.0081$ for L and for $1.5 \times 10^{-4}$ for (**M**). Renal (**N**) Na$^+$ clearance (Cl$_{Na}$) and (**O**) Li$^+$ clearance Cl$_{Li}$, and (**P**) Li$^+$ clearance to Na$^+$ clearance ratio (Cl$_{Li}$/Cl$_{Na}$) in Kir4.2$^{+/+}$ and Kir4.2$^{-/-}$ mice ($p = 0.047$). Animals were on normal diets for all panels. $N = 5$ per group for (**A**, **B**, **D–H** and **N–P**), 9 and 8 per group for (**C**), and $N = 10$ per group for (**I–M**). *$P < 0.05$ by Student's $t$ test. All tests were two-sided. Data presented as mean ± sem.

diuretic response was not apparent on a normal K$^+$ diet, MK2206 did increase urine volume on 0 K (Fig. 5g). MK2206 did not affect the urine Na$^+$ to K$^+$ ratio under either dietary condition suggesting the distal nephron is not the dominant site mediating our observation (Fig. 5h). Since we observed reduced pAKT abundance in Kir4.2$^{-/-}$ animals on a K$^+$ deficient diet, we next compared the MK2206 response of knockouts and control animals on 0 K. The natriuretic and diuretic responses trended towards being reduced in Kir4.2$^{-/-}$ animals, but did not achieve statistical significance (Fig. 5i, Supplementary Fig 8d). The kaliuretic response, however, was blunted in knockouts compared to controls (Fig. 5j). Strongly supporting a role for AKT signaling in the renal hypertrophic response to low K$^+$ was the absence of an increase in kidney mass in wild-type animals treated with MK-2206 while consuming a K$^+$-deficient diet for four days compared to vehicle-treated mice on an identical diet (Fig. 5k).

## Kir4.2$^{-/-}$ mice have reduced phosphorylation of mTOR targets on 0 K

Inhibition of kidney growth with both AKT inhibition and Kir4.2 deletion suggested a common mechanism. We detected mTOR staining along the PT as determined by immunofluorescent costaining with LTL (Fig. 6a). Kir4.2$^{-/-}$ animals on 0 K had reduced phosphorylated mTOR abundance compared with control mice (Fig. 6b, Supplementary Fig 9a, b). Knockouts also had reduced abundances of phosphorylated targets downstream of mTOR signaling including P70S6 kinase, ribosomal protein S6, eIF4G, and NDRG1 (Fig. 6c, d Supplementary Fig 9c–f). Total abundances of NBCe1 and NHE3 were also both reduced in Kir4.2$^{-/-}$ mice on 0 K compared to controls (Fig. 6e, Supplementary Fig 9g, h).

Further supporting reduced PT NHE3 function was a reduction in urine ammonia excretion in knockouts (Fig. 6f).

## Renal mTORC2/AKT signaling is activated via PT basolateral Cl$^-$ transport in response to reduced extracellular K$^+$

To determine the underlying mechanism by which low K$^+$ activates mTOR and AKT we moved to an ex vivo tubule suspension model. Reductions in extracellular K$^+$ are known to stimulate K$^+$-sensitive pathways in this model and we were able to confirm increased pNCC in this setting as a positive control (Supplementary Fig 10a)[23]. To determine if reduced extracellular K$^+$ itself is capable of increasing pAKT, we cultured tubule suspensions in either 6 mM K$^+$ or 0 mM K$^+$ for 30 min and found significant increases in pAKT while total AKT abundance remained unchanged (Fig. 7a). This effect was graded throughout the entire range of physiological extracellular K$^+$ (Fig. 7b). This effect was not sensitive to rapamycin, which predominantly inhibits the mTORC1 complex (Supplementary Fig 10b), but it was prevented with the mTORC2 inhibitor AZD8055 (Fig. 7c). We next observed low K$^+$-mediated phosphorylation of mTOR target proteins, including p70 S6 kinase, S6 ribosomal protein, and NDRG1 (Fig. 7d). These were the same phosphosites that were reduced in Kir4.2$^{-/-}$ animals on 0 K (Fig. 6). While activation of these targets is largely dependent on the mTORC1 complex, we investigated if there was activation of mTORC1 downstream of mTORC2/AKT. In support of this, we observed low K$^+$-dependent phosphorylation of mTOR itself and TSC2, which is known to mediate crosstalk between mTORC2 and mTORC1 signaling. AKT inhibition with MK2206 treatment in the tubule system prevented each of these phosphorylation events (Fig. 7d).

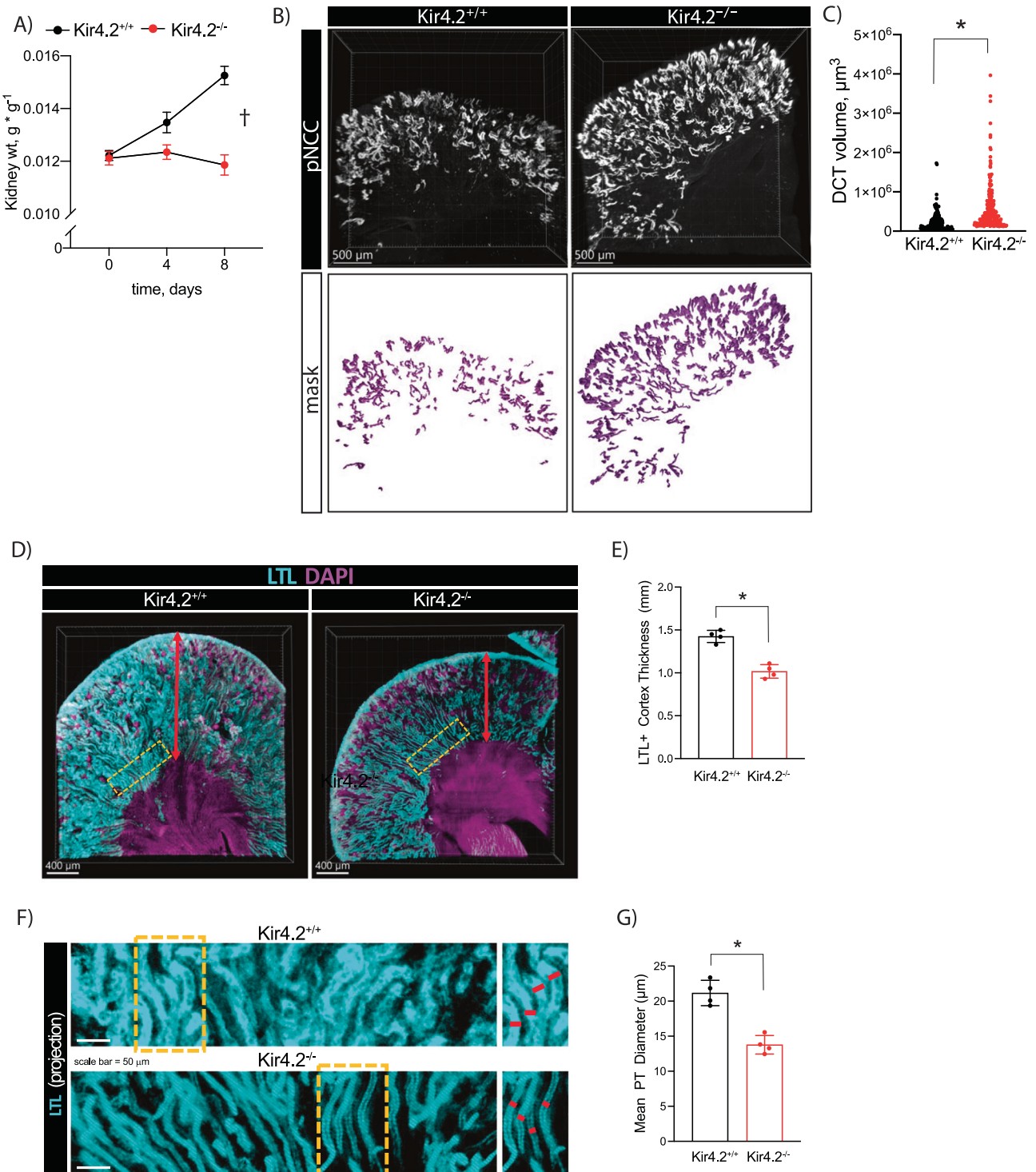

**Fig. 4 | Effects of Kir4.2 deletion on kidney growth and nephron segment expansion. A** Kidney mass from Kir4.2$^{+/+}$ and Kir4.2$^{-/-}$ mice on 0 K diet at indicated timepoints ($p < 0.0001$ for interaction). **B** Representative three-dimensional images from optically cleared Kir4.2$^{+/+}$ and Kir4.2$^{-/-}$ kidneys stained for pNCC-T53. Animals were maintained on 0 K diet for eight days. **C** Quantification of DCT volume based on imaging as described in B ($3.4 \times 10^{-10}$). **D** Representative three-dimensional images from optically cleared Kir4.2$^{+/+}$ and Kir4.2$^{-/-}$ kidneys stained with LTL-fluorescein and DAPI. Animals were maintained on 0 K diet for eight days. Yellow boxes indicate sections highlighted in (**F**). **E** Quantification of cortical thickness based on imaging as described in (**D**) ($p = 2.5 \times 10^{-4}$). **F** Representative 2-D projections from optically cleared Kir4.2$^{+/+}$ and Kir4.2$^{-/-}$ kidneys stained with LTL-fluorescein. Yellow boxes indicate sections highlighted on the right.
**G** Quantification of tubule thickness based on imaging as described in (**F**) ($p = 5.8 \times 10^{-4}$). In (**A**), $N = 10$ ($t = 0$), 5 ($t = 4$), and 11 ($t = 8$) for Kir4.2$^{+/+}$ and $N = 10$ ($t = 0$), 15 ($t = 4$), and 14 ($t = 8$) for Kir4.2$^{-/-}$, for (**B**–**G**), $N = 3$ per group $^{†}$P < 0.05 for interaction between treatment and time variables by two-way ANOVA with repeated measures. * indicates $P < 0.05$ by unpaired Student's $t$ test. All tests were two-sided. Data presented as mean ± sem.

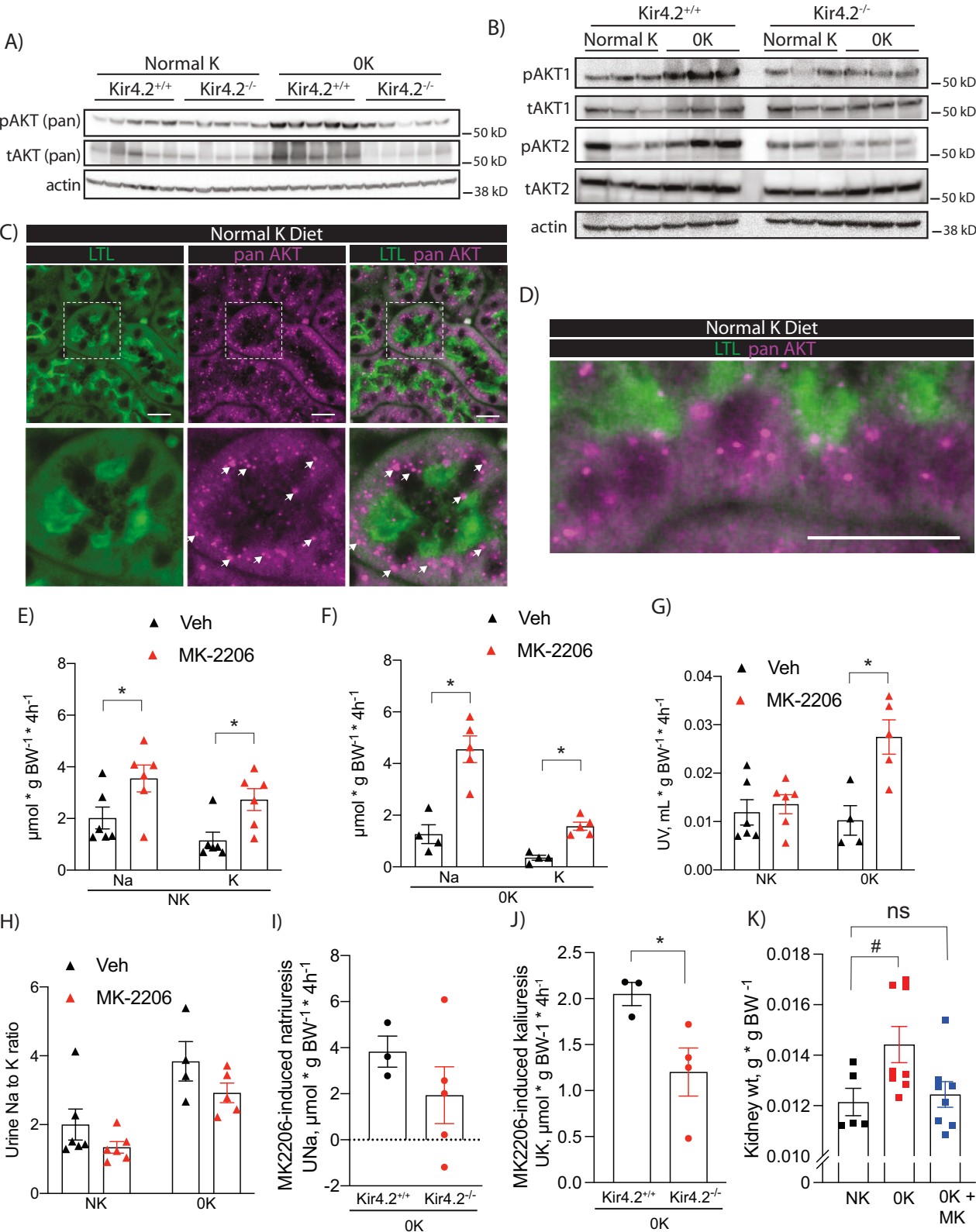

Potassium-induced phosphorylation events that occur along the distal nephron are mediated by changes in membrane voltage[5,7]. To determine if this was also the case for AKT, we treated cells with barium, which inhibits Kir channels, and found this completely prevented low $K^+$ increases in pAKT (Fig. 7e). Reducing extracellular $K^+$ also has effects on cytosolic pH, causing intracellular acidosis due to increased bicarbonate efflux[24]. To determine

if altering pH affects pAKT abundance, we cultured tubules in 4 mM $K^+$ while varying medium pH, and found this did not affect pAKT abundance (Fig. 7f). When we varied medium $HCO_3^-$ in the setting of stable pH and $K^+$, we similarly found that pAKT abundance was not affected (Fig. 7g).

Distally, low $K^+$-induced changes in membrane voltage influence phosphorylation events via changes in intracellular $Cl^-$ concentration[6].

**Fig. 5 | Effects of dietary K⁺ on AKT phosphorylation and function in the kidney.** **A** Representative Western blots for pan pAKT-S473 and pan total AKT from Kir4.2⁺/⁺ and Kir4.2⁻/⁻ kidneys following normal K⁺ and 0 K dietary treatments. Quantification presented in Supplementary Fig. 5. **B** Representative Western blots using isoform-specific antibodies for pAKT1, total AKT1, pAKT2, and total AKT2 from mice treated as in (**A**). Quantification presented in Supplementary Fig. 5. **C** and **D** Representative immunofluorescence imaging showing colocalization of AKT and the PT marker LTL in mice maintained on a normal diet. Arrows indicate punctate staining of AKT. Urinary Na⁺ and K⁺ excretion following treatment with either vehicle (Veh) or MK-2206 in wild-type animals on either (**E**) normal K⁺ (NK, $p = 0.046$ and 0.014) diet or (**F**) 0 K diet ($p = 0.0059$ and 0.0079). **G** Diuretic response ($p = 0.0092$) and (**H**) the urine Na-to-K ratio following to the same treatments as in (**D**). MK-2206-induced (**I**)

urine Na⁺ and (**J**) K+ response in Kir4.2⁺/⁺ and Kir4.2⁻/⁻ animals ($p = 0.048$). Each data point presented for (**I**) and (**J**) is the difference between MK-2206- and vehicle-induced electrolyte excretion for each animal. **K** Kidney weights from normal K-fed (NK) mice, 0K-fed (0 K) mice, and 0K-fed mice that were also treated with MK-2206 (0 K + MK) for four days ($p = 0.046$). $N = 5$ per group for all in (**A, C**), $N = 3$ per group for (**B**), $N = 6$ per group for (**E**), and $N = 4$ for Veh and 5 for MK-2206 for (**F**). In (**G**), $N = 6, 6, 4$ and 5. In (**H**), $N = 5, 6, 4,$ and 5. $N = 3$ for Kir4.2⁺/⁺ and 4 for Kir4.2⁻/⁻ in (**I**) and (**J**). For (**K**), $N = 5, 8,$ and 8 respectively. * indicates $p < 0.05$ by unpaired Student's $t$ test. # indicates $P < 0.05$ by one-way ANOVA followed by Dunnett's post-hoc test. Scale bars = 20 μM. All tests were two-sided. Data presented as mean ± sem.

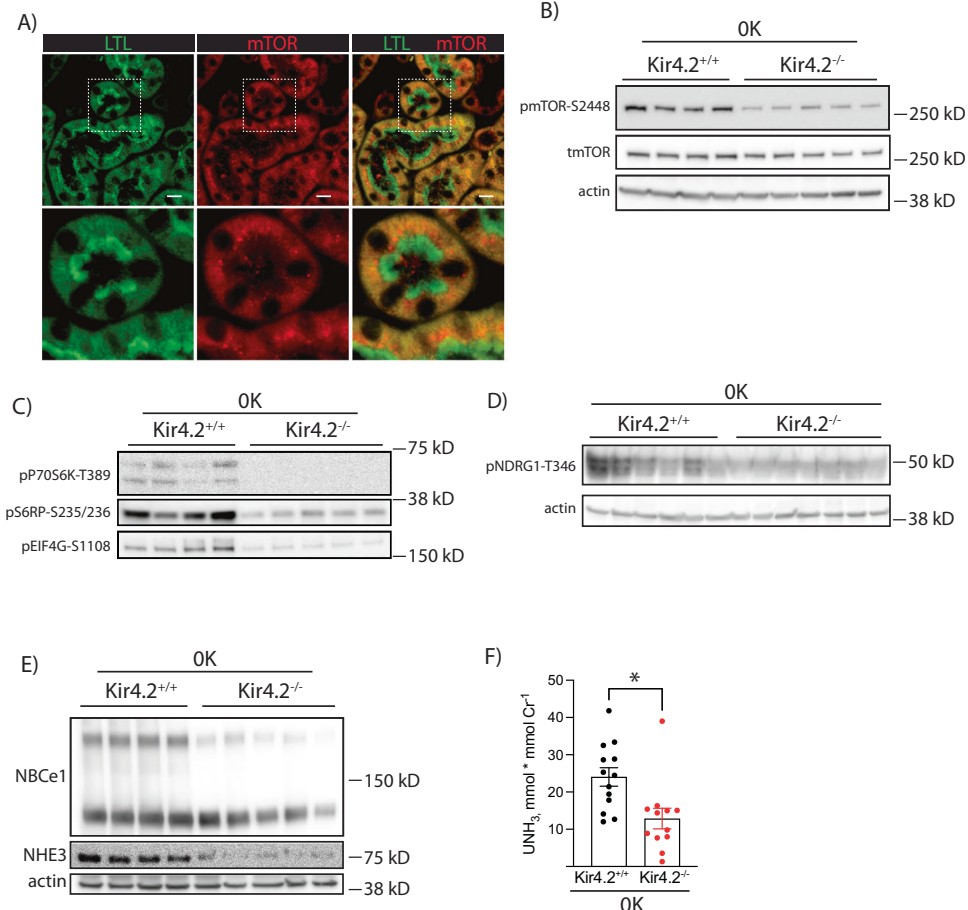

**Fig. 6 | Kir4.2 deletion reduces abundance of phosphorylated mTOR targets and PT Na⁺ transporters following 0 K feeding. A** Representative immunofluorescence staining for LTL and mTOR along the proximal tubule. **B** Representative Western blots for total mTOR and pmTOR-S2448 from Kir4.2⁺/⁺ and Kir4.2⁻/⁻ mice treated with 0 K. Quantification shown in Supplementary Fig. 4. **C** Representative Western blots for pP70S6 kinase-T389, pS6RP-S235/236, pEIG4G-S1108, and (**D**) pNDRG1-T346 from Kir4.2⁺/⁺ and Kir4.2⁻/⁻ mice treated with 0 K. Loading control for (**C**) is same as in (**B**). Quantification shown in Supplementary

Fig. 4. **E** Representative Western blots for NBCe1 and NHE3, from Kir4.2⁺/⁺ and Kir4.2⁻/⁻ mice treated with 0 K. Quantification shown in Supplementary Fig. 7. **F** Urine ammonia excretion from Kir4.2⁺/⁺ and Kir4.2⁻/⁻ mice treated with 0 K. 0 K treatment was given for 8 days for (**B–F**). For (**A**), $N = 5$ per group. For (**B, C, E**), $N = 4$ for Kir4.2⁺/⁺ and 5 for Kir4.2⁻/⁻. Panel (**D**) is a representative image from $N = 10$ and 12. For (**F**) $N = 13$ and 12. * indicates $P < 0.05$ by unpaired Student's $t$ test. Scale bar = 20 μM. All tests were two-sided. Data presented as mean ± sem.

Here we also found that reductions in medium Cl⁻ increased pAKT abundance in a 4 mM K⁺ bath (Fig. 7h). Raising medium Cl⁻ also prevented low K⁺-induced AKT phosphorylation (Fig. 7i). These data were consistent with a Cl⁻ efflux event reducing intracellular Cl⁻ concentration to stimulate mTORC2 and AKT. A recent report demonstrated the PT basolateral VRAC channel regulates PT function and its deletion in mice caused Fanconi syndrome[25]. To determine if low K⁺-induced Cl⁻ exit via PT basolateral VRAC channels, we treated cells with the VRAC inhibitor DCPIB in the setting of low K⁺ culture. DCPIB completely prevented increased pAKT observed in low K⁺ conditions (Fig. 7j).

The phosphorylation event was not prevented with the nonspecific Cl⁻-channel blocker, DIDS.

Data thus far were consistent with an effect of cell Cl⁻ modulating PT mTORC2/AKT activity; however, reduced cell volume in response to low extracellular K⁺ could also be contributing to kinase activation. The WNK kinases, while being modulated by Cl⁻, are also known to be molecular crowding sensors and are activated by cell shrinkage[26]. To test if kidney AKT is activated by reduced cell size, we treated tubules with a hypertonic stimulus to induce cell shrinkage. In contrast to effects of a 0 mM K⁺ bath, treatment with 50 mM mannitol did not alter

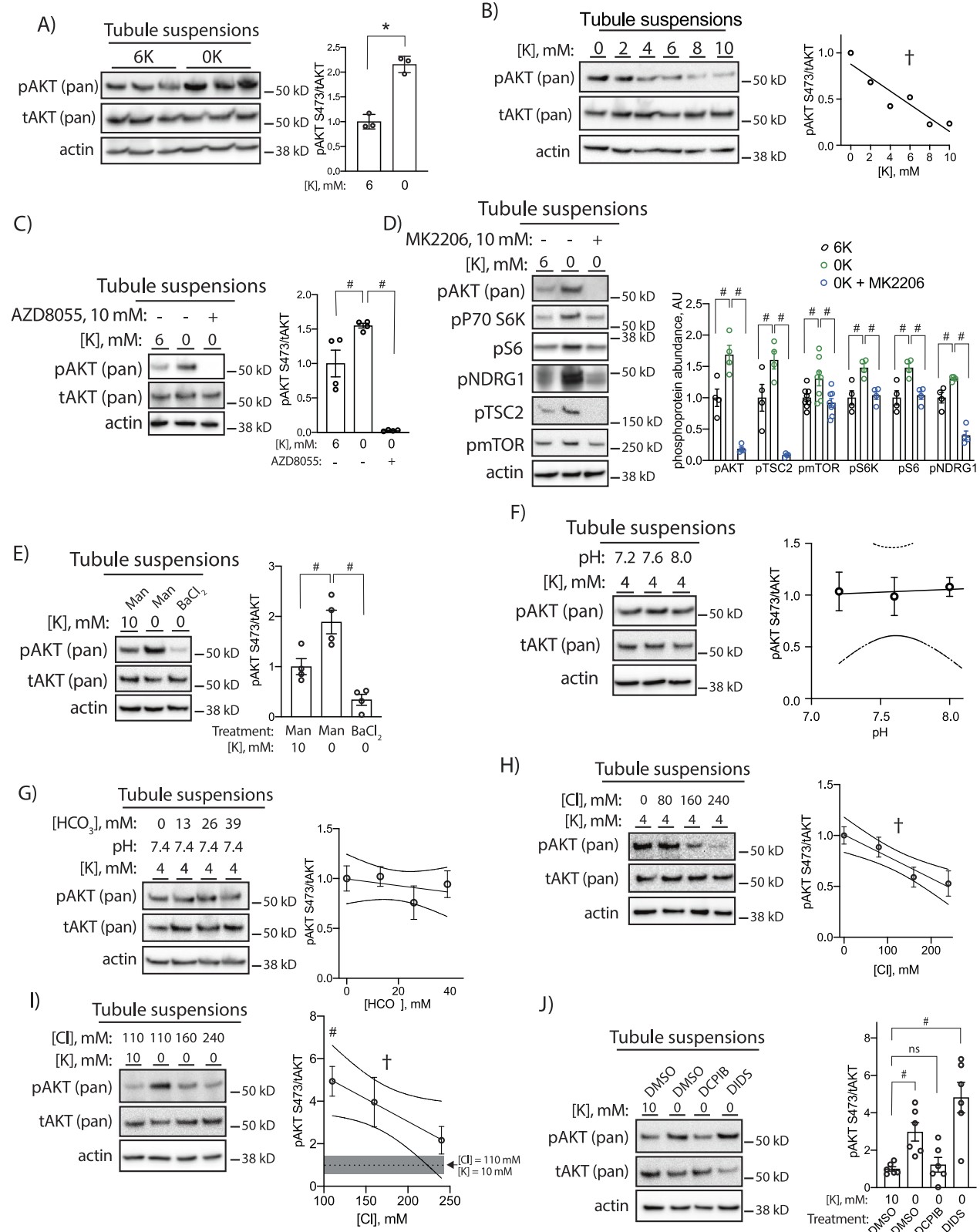

AKT phosphorylation (Fig. 8a). Similarly, treatment with 50 mM sorbitol did not affect phosphorylated AKT abundance, while hypertonic NaCl (additional 50 mM) tended to reduce pAKT (Fig. 8b).

## Discussion

The PT epithelium reabsorbs a majority of the filtered electrolyte load[27]. Both the PT and DCT increase Na$^+$ reabsorption and undergo hypertrophy in response to dietary K$^+$ depletion[1,4]. While the DCT has been intensely studied, mechanisms guiding the PT response have not been fully elucidated. Along the distal nephron, Kir4.1 mediates electrogenic basolateral K$^+$ efflux to coordinate extracellular K$^+$ changes with intracellular kinase signaling. Disruption of this signaling cascade at multiple points inhibits Na$^+$ reabsorption via NCC[5,28–31]. Here we identified Kir4.2 as serving a similar role along the PT to couple

**Fig. 7 | Effects of extracellular K+ reductions on isolated tubule suspensions.** **A** Western blots from isolated tubule suspensions cultured for 30 min in either 6 mM or 0 mM K+ conditions ($p = 8.5 \times 10^{-4}$). **B** Representative Western blots from isolated tubule suspensions cultured for 30 min in indicated K+ concentrations ($p = 0.008$ for nonzero slope). **C** Representative Western blots from isolated tubule suspensions cultured for 30 min in indicated K+ concentrations with or without the mTORC2 inhibitor AZD8055 (10 μM) ($p = 0.019$ and <0.0001). **D** Representative Western blots from isolated tubule suspensions cultured for 30 min in indicated K+ concentrations with or without the AKT inhibitor MK2206 (10 μM) ($p = 0.0069$ and <0.0001 for pAKT, $p = 0.044$ and 0.0001 for pTSC2, $p = 0.043$ and 0.0092 for pmTOR, $p < 0.0001$ and $p < 0.0001$ for pP70S6K, $p = 0.0078$ and 0.013 for pS6, and $p = 0.02$ and <0.0001 for pNDRG1). **E** Representative Western blots from isolated tubule suspensions cultured for 30 min in indicated K+ concentrations with medium supplemented with either mannitol (30 mM) or BaCl$_2$ (10 mM) ($p = 0.015$ and 0.0004). **F** Representative Western blots from isolated tubule suspensions cultured for 30 min in 4 mM K+ at indicated pH. **G** Representative Western blots from

isolated tubule suspensions cultured for 30 min in 4 mM K+ and pH 7.4 at indicated HCO$_3^-$ concentrations. **H** Representative Western blots from isolated tubule suspensions cultured for 30 min in 4 mM K+ at indicated Cl- concentrations ($p = 0.0009$ for nonzero slope). **I** Representative Western blots from isolated tubule suspensions cultured for 30 min in indicated K+ and Cl- concentrations ($p = 0.043$ for nonzero slope). **J** Representative Western blots from isolated tubule suspensions cultured for 30 min in indicated K+ concentrations and either DMSO, DCPIB (10 μM), or DIDS (100 μM) ($p = 0.036$ and 0.0001). $N = 3$ per group for (**A, F−I**), 4 per group for (**C, E**), 4 per group for (**D**) except pmTOR for which $N = 8$, and 6 per group for (**J**). * indicates $P < 0.05$ by unpaired Student's $t$ test. # indicates $P < 0.05$ by one-way ANOVA followed by Tukey's (**C−E**) or Dunnett's (**I, J**) post-hoc test. † indicates $P < 0.05$ for slope being significantly different than 0. In **I**, post-hoc comparisons are made between the indicated group and the 10 mM K+, 110 mM Cl- group, the abundance of which is indicated by a dotted line and SEM indicated by the shaded gray area. All tests were two-sided. Data presented as mean ± sem.

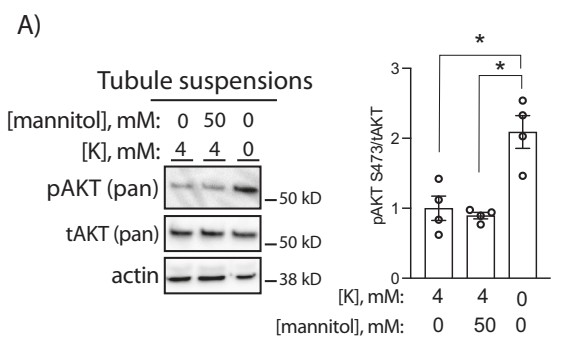

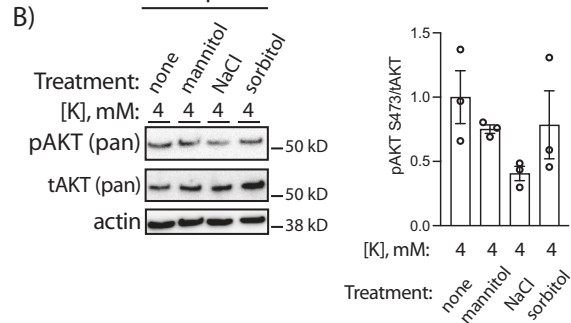

**Fig. 8 | Effects of extracellular hypertonicity on isolated tubule suspensions.** **A** Representative Western blots from isolated tubule suspensions cultured for 30 min in either 0 or 50 mM mannitol at indicated K+ concentrations. **B** Representative Western blots from isolated tubule suspensions cultured for 30 min at

50 mM concentrations of indicated solute. * indicates $P < 0.05$ by one-way ANOVA followed by Tukey's post-hoc test. $N = 4$ per group in A and 3 per group in (**B**). All tests were two-sided. Data presented as mean ± sem.

basolateral K+ changes with the intracellular response. Without Kir4.2, animals waste K+, develop a proximal RTA, and fail to undergo PT expansion compared to control animals. 0 K diet increased kidney AKT phosphorylation in Kir4.2$^{+/+}$ but not Kir4.2$^{-/-}$ mice. We mechanistically confirmed extracellular K+ reduction activates mTORC2 signaling in our ex vivo tubule culture system. Kir4.2 knockout animals had reduced phosphorylation of mTOR targets suggesting disrupted mTOR signaling underlies the lack of kidney growth and transport deficiencies in these animals. Using isolated tubules in culture we demonstrated increased basolateral PT Cl- transport activates mTORC2 signaling in response to reductions in ambient K+.

## Kir4.2 is essential for the physiological K+ response

Kir4.2 deletion abrogated an increase in gross kidney size in response to low K+ feeding. Optical clearing and three-dimensional confocal imaging revealed a lack of tubule expansion along the PT in Kir4.2$^{-/-}$ animals. Conversely, the hypertrophic response along the DCT was not only intact, but appeared exaggerated relative to Kir4.2$^{+/+}$ mice. The absent change in knockout kidney weight identifies the PT as the site for most of the renal hypertrophic response. While the DCT does undergo hypertrophy[21], this is at the expense of a reduction in CNT mass[32] leaving the distal nephron size as a whole unchanged in the low K+ setting. Along with an absence of PT expansion, knockouts failed to compensate physiologically for dietary K+ deficiency showing the increase in mass is coupled to changes in transport. They wasted urinary K+ leading to more severe hypokalemia and acidosis compared with controls. These findings highlight an essential PT role for Kir4.2 in identifying and responding to changes in blood K+ levels.

Kir4.2 forms functional heterotetramers with Kir5.1 in vitro[33] and evidence suggests this occurs in vivo along the PT[34]. While

human disease-causing Kir4.2 mutations have not been identified, Kir5.1 mutations were reported to cause hypokalemia, renal Na+ wasting, acid-base disturbances, and sensorineural deafness[35]. This syndrome closely resembles SeSAME/EAST syndrome[9,36], which is caused by mutations in Kir4.1 (which heterotetramerizes with Kir5.1 along the DCT), and includes a kidney phenotype resembling a Gitelman tubulopathy. A clinical difference between the two syndromes relates to acid-base disturbances; Kir4.1 mutations invariably cause alkalosis, secondary to NCC inhibition, while Kir5.1 mutations are more variable with a subset of patients presenting with acidosis while others have alkalosis. Heterogeneity likely relates to Kir5.1 serving distinct roles along the PT and DCT and whether patients have predominantly proximal or distal disruption of transport processes. The novel syndrome suggests intact Kir4.2 function, through its interactions with Kir5.1, is essential to maintain normal physiology in humans.

## Increased distal transport sensitizes Kir4.2$^{-/-}$ animals to clinically useful diuretics

At baseline, the absence of Kir4.2 resulted in reduced PT Na+ transport, but knockouts did not have major electrolyte or blood pressure derangements. Results indicate that compensation was achieved by shifting the Na+ reabsorptive burden distally as demonstrated by increased total NKCC2, phosphorylated NCC, and cleaved gamma ENaC abundances compared to Kir4.2$^{+/+}$ animals. In knockout animals we observed a tendency towards water and K+ wasting in addition to elevated aldosterone levels, which are known physiological stimuli that activate transport via NKCC2, NCC, and ENaC, respectively. Distal segments are known to be highly plastic and can greatly increase their capacity for Na+ transport[32,37,38]. However, this led to increased

ENaC-mediated K+ secretion in the low K+ state resulting in excessive urine losses and worsened hypokalemia in knockouts.

A consequence of distal transport activation is that Kir4.2−/− mice had increased responsiveness to the commonly used diuretics HCTZ and furosemide. This observation has potential implications for solving the problem of clinical diuretic resistance. The most clinically useful diuretics target distal segments. It has long been thought that inhibition of proximal transport has transient effects and is generally an ineffective approach to induce diuresis. However, the more recent use of sodium glucose transporter 2 (SGLT2) inhibitors and renewed interest in acetazolamide has some practitioners and physiologists rethinking this paradigm[39,40]. Our results suggest the idea that blockade of proximal transport, in conjunction with distal inhibition, may produce potent synergistic effects to maximize diuresis. Co-inhibition of the PT in combination with the distal nephron could be a powerful approach to offset diuretic resistance encountered while treating patients in need of volume control. Kir4.2, or related signaling pathways, are targets to consider.

### AKT signaling coordinates the renal low K+ response

We observed reduced AKT phosphorylation in Kir4.2 knockout animals on 0 K relative to controls. This prompted us to consider AKT in the epithelial response to low K+. Most studies have found AKT does not contribute significantly to the regulation of distal transport[14–16]. We found that the AKT inhibitor, MK2206, induced a potent natriuresis and kaliuresis and this response was enhanced under low K+ conditions consistent with a low K+ activation of AKT. The effects on Na+ and K+ excretion were comparable, leaving the urine Na+-to-K+ ratio unchanged. While our Kir4.2 knockout data suggest a compensatory distal activation of Na+ transport in Kir4.2−/− animals in the chronic setting, an unchanged urine Na+-to-K+ ratio with MK2206 suggests minimal effects on principal cell-mediated Na+ reabsorption acutely. Acute alterations in ENaC function would likely affect this ratio. Moreover, we did not observe increased kidney mass with AKT inhibition suggesting it plays an important role in low K+-mediated renal growth. PT expansion is a major contributor to increased kidney size, suggesting an important proximal role for AKT in this process. We found both AKT1 and AKT2 isoforms undergo low K+-induced phosphorylation while AKT3 was not detectable in kidney. AKT exhibited a punctate staining pattern under both normal and K+-deficient feeding suggesting localization of the kinase itself was not a key determinant of altered signaling under low K+ conditions. Further clarification of the roles of the individual isoforms and how potential binding partners augment AKT signaling under these conditions will require future investigation.

### The mTORC2 complex activates AKT in response to low K+ exposure

mTOR has been suggested to play a role along the PT[41,42], though this has not been investigated under low K+ conditions previously. Our ex vivo tubule system facilitated identification of mTORC2 as the complex stimulating AKT following low K+ exposure and we confirmed phosphorylation of multiple components of the mTOR signaling pathway. Crosstalk between the mTORC1 and 2 complexes via TSC2 appears to underlie phosphorylation of several key targets. The reduced abundance of mTOR targets in Kir4.2−/− animals suggests Kir4.2, a PT-specific channel, is required for the low K+-mediated activation of mTOR signaling along the PT.

A substantial body of work has focused on mTORC2 activation of SGK1 and ENaC in principal cells in response to high K+ exposure. Previous studies suggest cellular- and context-specific conditions determine the outcome of mTORC2 signaling[43]. mTORC complexes are multiunit structures that are regulated not only by phosphorylation, but also by specific subunit members, their assembly, and their localization. Our study presented here adds to the body of work suggesting mTORC2 plays distinct roles in different cell types and can be

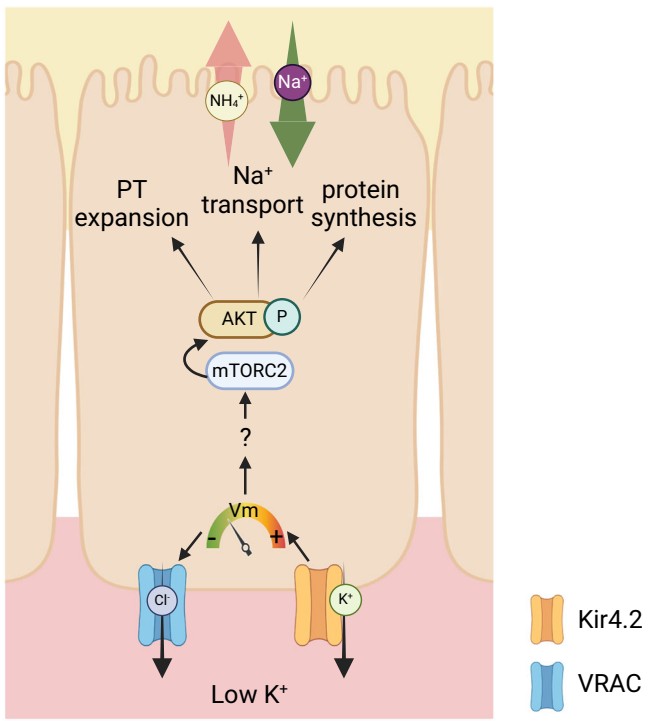

**Fig. 9 | Model depicting effects of reduced extracellular K+ on PT cell physiology.** Reduced blood K+ along the basolateral membrane leads to K+ efflux via Kir4.2, which reduces membrane voltage (Vm). This promotes Cl− efflux through VRAC, decreased intracellular Cl− concentration [Cl−], and activates mTORC2 to phosphorylate AKT. This has broad effects on cell physiology including increasing transcription/translation, Na+ reabsorption, and PT expansion. Figure 9 created with BioRender.com released under a Creative Commons Attribution-NonCommercial-NoDerivs 4.0 International license https://creativecommons.org/licenses/by-nc-nd/4.0/deed.en.

activated in a nephron-specific manner depending on the extracellular stimulus. (Fig. 9). A limitation is that we only use pharmacological inhibitors to draw these conclusions and not genetically altered animals; the generation of meticulous genetic animal models in the future will be required for a complete investigation of mTOR/AKT regulation along the PT.

### Intracellular Cl− is a K+-responsive ion coordinating the proximal and distal low K+ responses

Our ex vivo studies identified Cl− as a mediator of extracellular K+ changes and intracellular signaling. Cl− was first identified in a similar role to coordinate K+-sensitive intracellular signaling along the DCT[7] and then more recently along the CNT[44]. In our current case, the basolateral VRAC channel, which is known to regulate PT function[25], appeared to be facilitating changes in PT cell Cl−, similar to the ClC-K2 channel along the DCT[7,45]. A limitation of our study is that we were unable to measure [Cl−] in tubule cells directly. Transgenic animals expressing Cl-sensitive fluorescent probes have been proposed as a way to perform these measurements distally[46]. This is an attractive approach to perform such measurements in proximal epithelial cells in the future.

We also considered whether reduced cell volume, perhaps via molecular crowding, is contributing to AKT activation akin to a mechanism that has been described for WNK kinases[26]. Our results did not support a dominant role for reduced cell volume in our tubule system; however, our data cannot eliminate the possibility that mTOR/AKT is involved in the cellular response to cell volume changes more broadly. The ex vivo tubule system provides valuable mechanistic insight, but its restriction to acute timepoints is a limitation and additional studies will be required to determine if these current data

reflect chronic changes occurring in dietary models. While our data support a Cl⁻-sensitivity of mTORC2/AKT signaling, future studies will also be required to provide a detailed description of how reduced Cl⁻ affects the activity or the complex itself, or its upstream regulators. The repeated identification of cell Cl⁻ as an important regulator of intracellular signaling suggests a more widespread existence of this phenomenon, potentially in multiple organ systems, to regulate systemic physiology. In the kidney, it serves to coordinate the proximal and distal low K⁺ responses towards the preservation of renal K⁺ homeostasis.

## Methods

### Mice
All animal experiments were performed in accordance with the guidelines and with the approval of the Institutional Animal Care and Use Committee of Vanderbilt University Medical Center. Mice were used for experimentation were aged 8-12 weeks, on a C57Bl/6 background. Kir4.2 knockout mice were generated by J. Denton and reported previously[24]. All mice were genotyped by PCR on ear or tail snip DNA before and after experiments. Primers used for Kir4.2 knockout genotyping include forward: GGTAGGAGATTAACACCA TACTG and reverse: GAGAGTCCACTTTCATAATGCAG.

### Metabolic cage experiments
Animals were individually housed in metabolic cages from 1700 until 0800 the following day. When urine was collected for consecutive days as indicated, animals remained in their home cages between collections from 0800 until 1700 when they were returned to metabolic cages. Urine was collected under water-saturated light mineral oil. Urine Na⁺ and K⁺ were either measured immediately following collection on a Diamond Diagnostics Carelyte Plus analyzer (Holliston, Ma), or frozen until measurements were performed.

### Dietary/water treatments
Baseline data were collected from animals maintained on our standard animal diet (1.21% K⁺ and 0.99% NaCl). K⁺-deficient (TD.88239, 15-30 ppm K⁺, 0.3% Na⁺, and 0.45% Cl⁻, Envigo), NaCl-deficient (TD.90228, 0.8% K⁺, 0.01-0.02% Na⁺, 0.07% Cl⁻, Envigo), and K⁺ and Na⁺-deficient (TD.09841 15-30 ppm K⁺, 15-30 ppm Na⁺, and 0.45% Cl⁻, Envigo) diets were used where indicated. Amiloride was supplemented in drinking water at a concentration of 75 mg/L. Triamterene was supplemented in diet at a concentration of 100 mg/kg body weight.

### Diuretic response tests
For administration of hydrochlorothiazide (HCTZ, Sigma) and furosemide (Sigma), animals were gavaged with 25 mg/kg (dissolved in 3:1 PEG:H₂O). Following treatment animals were placed in metabolic cages and urine was collected for three hours. For MK2206, animals received an intraperitoneal (IP) injection at a dose of 50 mg/kg for all presented data except at 25 mg/kg for Supplementary fig. 4g (dissolved in 50% saline, 45% PEG, 5% DMSO). Following treatment animals were placed in metabolic cages and urine was collected for four hours. Urine Na⁺ and K⁺ were measured on a Diamond Diagnostics (Holliston, Ma) Carelyte Plus unit. Urine calcium was measure using the o-cresolphthalein complexone method (Pointe Scientific). Animals were treated with MK-2206 or vehicle (50% saline, 45% PEG, 5% DMSO) once per day (50 mg/kg, IP) for four consecutive days for Fig. 5k.

### Spot urine collection and electrolyte measurements
Spot urine was collected in the morning at indicated time points. Urine Na⁺ and K⁺ were measured on a Diamond Diagnostics Carelyte Plus analyzer (Holliston, Ma). Urine ammonia was measured after 1:50 dilution (Pointe Scientific) and normalized to creatinine that was measured with The Creatinine Companion kit (Exocell). Urine osmolality was measured on a Precision Systems freezing point osmometer

(6002). Free water clearance was calculated based on the formula $C_{H2O} = U_{vol} - [(U_{osm} \times U_{vol})/P_{osm}]$.

### Blood electrolyte measurements
Blood electrolytes were measured on samples obtained by cardiac puncture with an iSTAT analyzer using Chem8+ cartridges (Abbott, Abbot Park, Il) or with a Diamond Diagnostics Carelyte Plus analyzer (Holliston, Ma).

### Immunofluorescence Imaging
Staining on all sections embedded in paraffin (which included all 2D images except Supplementary fig. 5) weas performed as previously reported[47,48]. Briefly, following euthanasia kidneys were immediately placed in fixative as in prior reports and placed on rocker overnight at room temperature. Tissues were then dehydrated in a series of ethanols, embedded in paraffin, sectioned (5 μm), and mounted on glass slides for staining. Antibodies used are listed in Table 1. For each experiment, all kidneys were embedded in the same block and mounted on the same slide to reduce sample-to-sample variability. Images were imaged with a Nikon TE300 fluorescence microscope and spot-cam digital camera (Diagnostic Instruments). For 3D imaging presented in Supplementary fig. 5, kidneys were placed in 4% paraformaldehyde (PFA) immediately following euthanasia. Following an overnight incubation in PFA, kidneys were transferred to 30% sucrose for an overnight incubation. Kidneys were then embedded in OCT followed by sectioning on a cryostat (50 μm thick). Images were acquired with a Super Resolution Airy Scan 2 Detector.

### Optical clearing and analysis
Optical clearing was performed as previously described with some modifications[49]. Animals were euthanized and the kidney was removed, coronally sliced with a razor at 1–2 mm thickness, and slices were incubated for 30 min in freshly prepared 4% PFA rocking at room temperature, washed three times in PBS, and then incubated in Blocking buffer (2%BSA, 0.2% Tween, 0.2%, Triton-X100, 0.02% sodium azide) for 4–5 h at room temperature. Next, sections were incubated in primary antibody (at least 1:50 dilution depending on the primary) in blocking buffer for 3 days at room temperature on a rocking platform, washed, and then incubated in secondary antibodies for 3 days at room temperature. Slices were then washed thoroughly three times in PBS with 2-3 h per washing step. Slices were dehydrated in a methanol series (25%, 50%, 75%, 100%, 100%), 5 min each step, and transferred into BABB solution (Benzyl alcohol and benzyl benzoate at a 1:2 ratio) in an Eppendorf tube for 20 min. Once the slices were translucent, they were placed into a glass-bottom dish (Mattek #P35-1.5-14-C) containing 100 μL BABB solution for confocal imaging. Quantification of distal tubular volume was performed using Imaris (Bitplane) by automated 3D surface rendering around pNCC+ tubules. Data are presented for one representative animal from each genotype. Quantification of LTL+ positive cortex thickness was measured by creating maximum intensity projections of whole thickness kidney slices and using the line tool in ImageJ/FIJI. Proximal tubular width was determined by measuring the cross-sectional diameter of longitudinal maximum intensity projections of whole LTL+ tubules in ImageJ/FIJI.

### Lithium clearance
Animal diet was supplemented with LiCl at a dose of 20 mmol/kg diet. Animals were fed this diet for two days prior to urine collection in metabolic cages from 1700 until 0800 followed by euthanasia. Urine volume was determined and blood and urine Li⁺ concentrations were determined using Infinity Liquid Lithium Stable Reagent (Thermo Fisher) according to a previous report[50]. Lithium clearance was calculated using the formula: $Cl_{Li} = (U_{Li} \times V)/P_{Li}$ where $Cl_{Li}$ is the lithium clearance, $U_{Li}$ is the lithium concentration in collected urine, $V$ is the urine volume over collection period, and $P_{Li}$ is the lithium

## Table 1 | Antibodies

| Antigen | Host | Dilution | Source | Catalog/reference | Application |
|---|---|---|---|---|---|
| Alpha ENaC | Rabbit | 1:1000 | Stressmarq | SPC-403 | WB |
| Gamma ENaC | Rabbit | 1:1000 | Stressmarq | SPC-405D | WB |
| pNCC-T53 | Sheep | 1:1000 (WB) | Phosphosolutions | p1311-53 | WB |
| Total NCC | Rabbit | 1:10,000 | D. Ellison | 37 | WB |
| Total NKCC2 | Sheep | 1:1000 | MRC Dundee | S838B | WB |
| Kir4.2 | Rabbit | 1:1000 | Alomone Labs | APC-058 | WB |
| NHE3 | Rabbit | 1:1000 (WB) | Stressmarq | SPC-400 | WB |
| NBCe1 | Rabbit | 1:1000 | Abcam | SPC-400 | WB |
| LTL-fluorescein | | 1:50 | Vector | FL-1321-2 | IF |
| Pan pAKT-S473 | Rabbit | 1:1000 (WB) | Cell signaling | 4060[52], | WB |
| Total pan AKT | Rabbit | 1:1000; 1:50 (IF) | Cell signaling | 4691 | WB, IF |
| pAKT1-S473 | Rabbit | 1:1000 | Cell signaling | 9018 | WB |
| Total AKT1 | Rabbit | 1:1000; 1:50 (IF) | Cell signaling | 2938 | WB, IF |
| pAKT2-S474 | Rabbit | 1:1000 | Cell signaling | 8599[52], | WB |
| Total AKT2 | Rabbit | 1:1000; 1:50 (IF) | Cell signaling | 3063[52], | WB, IF |
| Total AKT3 | Rabbit | 1:1000 | Cell signaling | 4059 | WB |
| pmTOR S2448 | Rabbit | 1:1000 | Cell signaling | 5536 | WB |
| Total mTOR | Rabbit | 1:1000 (WB); 1:50 (IF) | Cell signaling | 2983 | WB, IF |
| pTSC2-T1462 | Rabbit | 1:1000 | Cell signaling | 3617 | WB |
| pP70 S6 kinase - T389 | Rabbit | 1:1000 | Cell signaling | 9205 | WB |
| pS6 S235/236 | Rabbit | 1:1000 | Cell signaling | 2211 | WB |
| pEIF4g S1108 | Rabbit | 1:1000 | Cell signaling | 2441 | WB |
| pNDRG1 T346 | Rabbit | 1:1000 | Cell signaling | 5482 | WB |
| Kir4.1 | Rabbit | 1:1000 | Alomone Labs | APC-035 | WB |
| Na/K ATPase | Rabbit | 1:50 | Invitrogen | ST0533 | IF |
| Actin | Mouse | 1:3000 | Millipore | A1978 | WB |

concentration in plasma. Sodium clearance ($C_{Na}$) was calculated in a similar fashion using urine and plasma sodium concentrations measured on a Diamond Diagnostics Carelyte Plus unit (Holliston, Ma).

### Invasive blood pressure measurement

Mice underwent surgery at the Vanderbilt Mouse Metabolic Phenotyping Center to implant a carotid artery catheter 4 days before blood pressure measurements as previously described[51]. Briefly, mice were anesthetized with isoflurane, and the left common carotid artery was catheterized for sampling using a 1Fr-2Fr transition polyurethane catheter (Instech Laboratories). The free catheter end was tunneled under the skin to the back of the neck and attached to a vascular access button (Instech Laboratories). Blood pressure data was collected every minute for 30 min on days four to six after surgery. Reported data is the average of the last 15 minutes of the sampling period after blood pressures had stabilized following tubing connection. The average of the three days were averaged to provide the reported values.

### Blood pressure measurement by tail cuff

Blood pressure was measured by tail cuff using a BP-2000 Series II blood pressure analysis system (Visitech systems). Animals were trained on the machine for a minimum of three days during which data was not recorded. Blood pressure was then determined for two consecutive days.

### Tubule suspension isolation and culture

Based on a previous report[23], kidneys were minced on ice for 30–60 s with a blade. Homogenates were then incubated in 1 mL RPMI medium

supplemented with collagenase D (Roche) and DNAse I (Biorad) for 5 min at 37 °C with intermittent agitation. Sample was then pipetted up and down 5-10 times and larger pieces were allowed to settle to the bottom. Overlying tubule suspension was removed and transferred to a new tube. An additional 1 mL of fresh medium with enzymes was added to remaining larger kidney pieces and incubation was repeated on both tubes for 5 min at 37 °C. Pipet agitation was performed again and tubule suspensions were removed from both tubes following settling of larger pieces. Suspensions were combined, resuspended in DMEM with 10% FBS, and centrifuged for 10 min at 300$g$ in a tabletop centrifuge. Tubules were plated in DMEM with 10% FBS for one hour in a 37 °C incubator with a $CO_2$ buffering system (5% $CO_2$) prior to use. After one hour, tubules were collected and centrifuged again for 10 min at 300$g$. After aspiration of supernatant, pellet was resuspended in indicated condition and treated for 30 min prior to collection.

Each tubule suspension treatment was performed for 30 min in a 37 °C cell culture incubator with a $CO_2$ buffering system (5% $CO_2$). Medium pH was adjusted to 7.4 prior to use unless otherwise noted. $K^+$-free DMEM was made in our lab according to previous report[7]. $K^+$ gluconate was used to supplement this base medium to achieve indicated $K^+$ concentrations. Rapamycin (Sigma), AZD8055 (MedChemExpress), and MK2206 (MedChemExpress) were all used at a final concentration of 10 µM. Mannitol and $BaCl_2$ were supplemented at final concentrations of 30 mM and 10 mM respectively. Experiments requiring pH adjustments were performed with HCl/NaOH. Bicarbonate-free medium was made using HEPES buffer and $NaHCO_3$ was added to achieve indicated $HCO_3^-$ concentrations. Na gluconate was used to control for $Na^+$ concentrations and tonicity. To modify

medium Cl⁻, a NaCl-free base medium was made to which either NaCl or Na gluconate was added back to control for Cl⁻ concentration with stable tonicity and Na⁺ concentration. 4-(2-Butyl-6,7-dichloro-2-cyclo-pentyl-indan-1-on-5-yl) oxobutyric acid (DCPIB, Tocris) and 4,4'- Dii-sothiocyano2,2'-stilbenedisulfonic acid (DIDS, Tocris) were used at final concentrations of 10 μM and 100 μM respectively.

Following treatment, suspensions were collected and pelleted by centrifugation in a microfuge for 5 min at 300$g$ followed by resus-pension in 1x sample buffer. Samples were then heated for 10 min at 65 °C followed by gel electrophoresis and Western blot.

## Western blot
Kidneys were snap frozen in liquid nitrogen at the time of euthanasia and subsequently transferred to −80 °C for storage. For tissue lysate preparation, tissue was homogenized with a Tissue-Tearor homo-genizer (Biospec Products) in lysis buffer as previously reported[47]. Lysate was then centrifuged at 3300$g$ for 15 min. Protein concentration was measured by BCA protein assay (Pierce) followed by gel electro-phoresis on a 4−20% Bis−Tris gel (Bio-Rad).

## Antibodies
Antibodies used are listed in Table 1.

## Statistical analysis
Data are presented as mean ± SEM. Statistical tests used are indicated in figure legends. Note for Fig. 2a one blood K⁺ value was excluded for the presence of gross hemolysis. Comparisons were made with unpaired Student's $t$ test or one/two-way ANOVA with or without repeated measures as appropriate and indicated in the figure legends. Corrections for multiple comparisons were performed as indicated. All tests were two-sided.

## Reporting summary
Further information on research design is available in the Nature Portfolio Reporting Summary linked to this article.

# Data availability
No large datasets conducive to deposition in a public repository were presented in this paper. Data reported in this paper will be made available by the corresponding author upon request. Source data are provided within this paper. Source data are provided with this paper.

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

## Acknowledgements

These studies were supported by NIH grants DP5-OD033412 (AST), DK51265, DK95785, DK62794, DK7569, P30DK114809 (R.C.H., M.Z.Z.), K08-DK135931 (J.P.A.), VA Merit Award 00507969 (RCH), and the Vanderbilt Center for Kidney Disease and Vanderbilt Diabetes Research and Training Center (DK020593) pilot and feasibility grant (J.S.D.). AMW was supported by NIH RO1-DK29857. A.S.T. was supported by a postdoctoral fellowship from the American Heart Association. F.B. was supported by a K08 (DK134879), a postdoctoral fellowship from the American Society of Nephrology, and a Vanderbilt Faculty Research Scholars Award. J.P.A. is a Robert Wood Johnson Foundation Harold Amos Medical Faculty Development Program Scholar. Microscopy was performed using the Vanderbilt Cell Imaging Shared Resource (supported by NIH grant P30-CA068485 and the Department of Veteran Affairs). The authors would like to acknowledge Dr. Eric Figueroa for assistance with generation of the Kir4.2 knockout animals and David H. Ellison for providing the total NCC antibody used in these studies and for critical reading of this manuscript.

## Author contributions

Y.Z., F.B., M.F., J.P.A., K.L.R., P.P., and A.S.T. contributed to acquisition, analysis, and interpretation of data. J.S.D., E.D., A.M.W., M.Z.Z., R.C.H., and A.S.T. contributed to conception and design. A.S.T. drafted the work and A.S.T., J.P.A., E.D., and R.C.H. substantively revised it. All authors approved of the final version.

## Competing interests

The authors declare no competing interests.
