## [Peer Review File · Nature Communications]

Low potassium activation of proximal mTOR/AKT signaling is mediated by Kir4.2REVIEWER COMMENTS

Reviewer #1 (Remarks to the Author):

This current paper is a followup to a 2022 paper to determine whether Kir4.2 mediates the physiological response to low K⁺. The authors previously showed that K⁺ deficiency causes kidney injury and Kir4.2 KO mice are protected from low K⁺ injury. Now they report that Kir4.2 mediates the proximal tubule response to low K⁺ through mTORC2/AKT → mTORC1 signaling. Overall this project has exciting findings, well designed experiments, and new insights into hypokalemic nephropathy. Some highlights include the quality of animal work, kidney imaging and quantification, and phosphoproteomics study. However, at times the story feels a bit disconnected between the first section about distal tubule Na⁺ reabsorption in Kir 4.2 KO mice, and the second part about mTOR signaling during low K⁺. The front section is ancillary to the main findings in the manuscript. The second section on mTOR was the stronger section.

1) The authors have done a very nice job linking modulation of chloride flux to mTORC2-Akt → mTORC1 activity. Though PT AKT and mTOR signaling has been studied before, this finding is new and significant. The major issue with the manuscript, however, is a lack of mechanism connecting cell chloride to mTORC2/Akt signaling and the overstated importance of chloride ion as a signal transducer. In the previous oft-referred to study evaluating effects of low K⁺ on distal tubule signaling, it was proposed that a kinase that directly and specifically senses Cl⁻, WNK4, mediates intracellular signaling during K⁺ deficiency. This chloride sensing function has been structurally resolved. mTORC2 has no such chloride ion-sensing function, yet it is proposed that Cl⁻ itself acts as an intracellular signaling molecule that inhibits mTORC2 in PT. This is not supported by the available data. It is not clear if intracellular chloride concentrations are changing significantly, as they were not measured. Measuring intracellular chloride seems essential to support the claim that the mechanism is analogous to distal tubule (realizing that the proposed [Cl⁻]_i mechanism in DCT has not been reproduced by all mouse models of WNK4 chloride-sensing inactivation). Even if such data were available, ideally, a clear mechanism connecting the effects of chloride to kinase activation should be resolved. In the absence of this, the emphasis on chloride as a bona fide signaling molecule throughout should be toned down and other potential explanations should be considered. For example, it is surprising that the authors have not considered the possibility that cytoplasmic crowding (cell shrinkage) rather than chloride ion may be the true signal that is regulating this process. Acute chloride and potassium exit from cells will reduce cell volume and thus increase local cytoplasmic macromolecular crowding. Blocking LRRC8/VRAC, a key mediator of RVD, will mitigate cell shrinkage and crowding, potentially explaining the effects of DCPIB. mTORC1 (downstream of mTORC2-Akt signaling here), is strongly regulated by cytoplasmic crowding, and tunes crowding via its effects on ribosomal biogenesis and cell size scaling (hypertrophy). Many mTORC1 processes that will influence scaling were uncovered via the nice differential phosphoproteomics analysis. Indeed, even the WNK kinases are also known direct crowding sensors and their activation mechanisms in DCT go well beyond pure chloride-binding

effects. It seems a consideration of cell volume/crowding changes in the PT should at least be considered and discussed.

2) Fig 1&2. Overall, the first section could be more mechanistic. The results suggest that ENaC activation is driving the increase in blood Na in KO mice, and the results with amiloride treatment correcting blood Na⁺ are convincing (2F). But how? Does amiloride increase urine Na excretion without affecting urine volume (not typically)? Blood Na⁺ can also increase due to increases in free water clearance. What are the urine volumes and osms in these mice? How does it change with time and diet. Why do the KO mice have an increase in UNa (1I) while also having an increase in blood Na⁺ (Fig 1B)? For Fig 1 it would be useful to show both the concentration of Na⁺ and the total Na⁺ based on urine volume. A limitation for the spot urines is that total Na⁺ cannot be determined. For Fig 2 what was UNa/Cr with amiloride treatment.

3) Fig 3: It's not clear to me what is driving the increase in ENaC, pNCC, and tNKCC2 activity. If the Kir4.2 KO mice have a trend towards lower serum K⁺ on control diet that could increase pNCC, but not ENaC and tNKCC2. This seems short on mechanism.

4) Fig 5C: Which AKT antibody was used for this? Is the punctate staining similar for AKT1 and AKT2? What does AKT staining look like in the Kir 4.2 KO mouse on control and low K⁺? If the signal is similar, what does it mean?

5) Fig 5 D & E, Sup Fig 3G & H: Are these figures showing the same data from the same mice? If so then what do the values differ for for urine K⁺ in WT mice fed 0K. The MK-2206 mice in Fig 5 have a max value ~2umol, whereas the MK-2206 mice in Sup Fig 3 have a max value of ~4.5umol.

6) In the first section the authors state that distal transport pathways compensate for proximal Na⁺ losses in Kir4.2 KO mice. In the second section the authors state that urine Na⁺ losses with MK-2206 treatment are not dependent on the distal pathway based on Fig 5G. However, this isn't clear why the distal pathway would be involved in the first instance and not the second.

7) Based on the NIH new Data Management and Sharing policy, please include details re: where the results from the phosphoproteomics study will be shared.

Minor:

1) line 320: "Large differences in blood electrolytes were not detected...consumptions of Na⁺ deficient diet". Yet, significant differences were seen in blood [Cl⁻] so this statement is not accurate. Please clarify.

2) line 330: "While 15-hour urine Na⁺ excretion was not different" Fig 1H. Is the 15h urine the amount of time it is collected? Please clarify what you mean by 15h since the X-axis is in days. Perhaps showing a treatment timeline would be helpful.

3) Fig 3: please indicate the diet these mice were on

4) Fig 3K: please switch lasix to furosemide

5) Sup Fig 3E. Can you include a positive control for tAKT3 to show that the antibody works? Or cite Mark Kneppers paper PMID: 33769951, showing very low levels of AKT3 mRNA that supports your finding.

6) Fig 5C: please indicate on the figure/image the type of diet used

7) Fig Sup 3F: please include a higher magnification insert as the punctate cannot be appreciated in this version

8) Fig Sup 3G: what is meant by "urine Na⁺ normalized to excretion", excretion of what?

9) Fig 5H: is this showing the concentration of Na⁺ per gBW at 4h? If so, why is there a negative value. Is it showing a change over 4h? Please clarify in the figure legend.

Reviewer #2 (Remarks to the Author):

This paper is an interesting extension of the authors previous description of the Kir4.2. knockout (previously published in Cell reports 2022). The authors aim to demonstrate that potassium homeostasis is altered. Unfortunately, the paper makes several claims that are not supported by the mechanistic data presented. The mechanistic data largely relies on ex vivo tubules that are treated with very large and non-physiological concentrations of ions, and on an AKT inhibitor.

Comments:

- Background and sexual dimorphism, if any, of the mice is not reported.
 - In fig. 2 they show that the excretion is dependent on ENaC, unfortunately only in short-term experiments and only using amiloride, an unspecific inhibitor.
 - In Fig. 3, the phosphorylation abundance (i.e. p-NCC) is regulated in the quantification, but not in the corresponding immunoblot. More careful immunoblot quantification is needed. Figure 3: What age did the mouse have?
 - In Fig. 4, the LTL projections are not very useful or very clear, since the LTL seems to label different parts of the cell – would this be also visible in normal EM/proximal tubule histology?
 - Figure 5C: The IF is of low quality. Please improve staining quality.
 - For instance, a human received 60mg, but the
 - Understanding blood pressure and activation of the renin-angiotensin-aldosterone systems, particular renin and aldosterone, are important to understand and support the mechanisms postulated here. Unfortunately, the blood pressure measurements are of low quality, because training was only done for 3 days (Suppl Fig. 1), and tail cuffs instead of telemetry were used.
 - Phosphoproteomics results cannot be interpreted without proteomics data. I.e. a phosphorylation switch might be real, or just a result of altered protein abundance. Data on proteome level, at least for representative phosphorylation sites important for nutrient sensing, should be provided. In addition using top 10 percentile is really not appropriate. One should use statistics with corrected false-discovery rates to make sure that these regulations are real. Similarly, the analysis was done using IPA analysis on protein level – not on the site level. It is not known if any of the regulated sites are functional, and bona-fide enriched kinase subsets for Akt/mTOR.
 - AKT function: The authors are not able to prove involvement of Akt in vivo. A high dose of the M2260 inhibitor (probably 100x of the human dose) is used, likely causing off-target effects. The AKT inhibitor M2260 would expect to alter blood glucose, which might change proximal tubule function, or diarrhea which would alter electrolyte balance.
- <https://ascopubs.org/doi/10.1200/JCO.2011.35.5263> Additional inducible genetic models are likely necessary to show involvement of AKT.

- Figure 7: mTOR activity is also regulated by phosphorylation, but particularly also by interaction as mTORC1 and mTORC2. This information on mTOR is missing, and would likely have to be tested by additional genetic studies. Like this, the altered phosphorylation remains an association, not a proof.

- Figure 8: Virtually all pH and ion concentrations are not in physiological range and thus, the regulations shown here are likely not physiologically relevant.

Minor comments

- Introduction: Kir 4.2, 4.1. Kir4.2. is a “nickname” for the real gene Kcnj15. To avoid ambiguity, please introduce correct gene names at least once in the paper.

- L90: “We previously reported that deletion of Kir4.2 prevents kidney injury caused by reductions in systemic K⁺ levels, but whether or not this channel mediates the physiological response to low K⁺ is unknown.” I think this is not entirely correct, since their cell reports paper already shows causal relationship between potassium, proximal tubule function and Kir4.2?

Reviewer #3 (Remarks to the Author):

The maintaining of systemic K⁺ levels is strongly associated with renal hypertrophy. Therefore, the identification of mediators and characterization of mechanisms behind hypokalemia-linked hypertrophy is crucial in preventing salt-sensitive hypertension. Previous studies established the involvement of distal tubule in salt-sensitive hypertension via distal activation of Na⁺ transport in response to inwardly rectifying K⁺ channel, Kir4.1 mediated low K⁺ states. In this elegant study, the authors identified Kir4.2 as a mediator of the proximal tubule response to K⁺ deficiency. The authors observed that removal of K⁺ from diet developed proximal renal tubular acidosis in knockout animals and increased urinary K⁺ excretion caused by ENaC-mediated increased distal sodium delivery. Kir4.2 has a role in proximal tubule expansion through mTORC2/AKT signaling. Moreover, they recognized intracellular Cl⁻ as a second messenger activating mTORC2/AKT mTORC2/AKT to coordinate cell growth and electrolyte transport in the low K⁺ response.

This study is comprehensive and exciting, and the findings are novel. The study should have significant value in understanding Kir4.2-mediated intracellular proximal ions homeostasis in regulating salt-sensitive hypertension. The experiments are well-designed and described. I have several comments, as listed below:

1. As Kir4.1 mediates electrogenic basolateral K⁺ efflux to coordinate extracellular K⁺ changes, what is the status of Kir4.1 in Kir4.2^{-/-} mice? Does the abundance of Kir4.1 increase when Kir4.2 is deleted?
2. Why does Na⁺ deficient diet have lower Cl⁻ (0.07%) than K⁺-deficient (0.45% Cl⁻)?
3. 'Renal mTORC2/AKT signaling is activated via changes in cytosolic Cl⁻ in response to reduced extracellular K⁺.' Is the Cl⁻ compartmentalized or cytosolic? It is more convenient to state intracellular Cl⁻ instead of cytosolic Cl⁻. What is the critical potassium level in the kidney that could cause Cl⁻-induced mTOR activation? What if the intracellular K⁺ and Cl⁻ levels in Kir4.2^{+/+} and Kir4.2^{-/-} mice were on a normal diet and 0K diet?
4. Fig. 3: Phospho and total NCC bands are not clearly visible, while others found prominent bands for these proteins (PMID: 36719746). Replace these blots with more visible bands.
5. Fig. 7C: The pNDRG1 band is not clear. Replace the blot.

Minor comments:

1. Supplemental Figure 5: Western blot quantification for figure 8b-d. But the quantifications are for Figure 7b-d.
2. Fig4A: 'Kidney mass from Kir4.2^{+/+} and Kir4.2^{-/-} mice on a normal diet and 0K diets at indicated timepoints.' However, the data showed only one plot for WT and KO. According to the result section description, it seems to be a 0K diet. Need clarification.

Reviewer #1 (Remarks to the Author)

This current paper is a followup to a 2022 paper to determine whether Kir4.2 mediates the physiological response to low K⁺. The authors previously showed that K⁺ deficiency causes kidney injury and Kir4.2 KO mice are protected from low K⁺ injury. Now they report that Kir4.2 mediates the proximal tubule response to low K⁺ through mTORC2/AKT → mTORC1 signaling. Overall this project has exciting findings, well designed experiments, and new insights into hypokalemic nephropathy. Some highlights include the quality of animal work, kidney imaging and quantification, and phosphoproteomics study. However, at times the story feels a bit disconnected between the first section about distal tubule Na⁺ reabsorption in Kir 4.2 KO mice, and the second part about mTOR signaling during low K⁺. The front section is ancillary to the main findings in the manuscript. The second section on mTOR was the stronger section.

1) The authors have done a very nice job linking modulation of chloride flux to mTORC2-Akt → mTORC1 activity. Though PT AKT and mTOR signaling has been studied before, this finding is new and significant. The major issue with the manuscript, however, is a lack of mechanism connecting cell chloride to mTORC2/Akt signaling and the overstated importance of chloride ion as a signal transducer. In the previous off-referred to study evaluating effects of low K⁺ on distal tubule signaling, it was proposed that a kinase that directly and specifically senses Cl⁻, WNK4, mediates intracellular signaling during K⁺ deficiency. This chloride sensing function has been structurally resolved. mTORC2 has no such chloride ion-sensing function, yet it is proposed that Cl⁻ itself acts as an intracellular signaling molecule that inhibits mTORC2 in PT. This is not supported by the available data. It is not clear if intracellular chloride concentrations are changing significantly, as they were not measured. Measuring intracellular chloride seems essential to support the claim that the mechanism is analogous to distal tubule (realizing that the proposed [Cl⁻]_i mechanism in DCT has not been reproduced by all mouse models of WNK4 chloride-sensing inactivation). Even if such data were available, ideally, a clear mechanism connecting the effects of chloride to kinase activation should be resolved. In the absence of this, the emphasis on chloride as a bona fide signaling molecule throughout should be toned down and other potential explanations should be considered. For example, it is surprising that the authors have not considered the possibility that cytoplasmic crowding (cell shrinkage) rather than chloride ion may be the true signal that is regulating this process. Acute chloride and potassium exit from cells will reduce cell volume and thus increase local cytoplasmic macromolecular crowding. Blocking LRRC8/VRAC, a key mediator of RVD, will mitigate cell shrinkage and crowding, potentially explaining the effects of DCPIB. mTORC1 (downstream of mTORC2-Akt signaling here), is strongly regulated by cytoplasmic crowding, and tunes crowding via its effects on ribosomal biogenesis and cell size scaling (hypertrophy). Many mTORC1 processes that will influence scaling were uncovered via the nice differential phosphoproteomics analysis. Indeed, even the WNK kinases are also known direct crowding sensors and their activation mechanisms in DCT go well beyond pure chloride-binding effects. It seems a consideration of cell volume/crowding changes in the PT should at least be considered and discussed.

We appreciate this reviewer's insightful comments here and throughout their review. We agree that cell volume itself, perhaps via molecular crowding, could be a critical mediator of the cell response to low K exposure. We have performed additional tubule experiments to test this hypothesis. These data are included as figure 9 in the revised version. We tested effects of hypertonic stimuli, including mannitol, sorbitol, and NaCl, on AKT phosphorylation. None of the

stimuli increased pAKT as low K did, suggesting that cell shrinkage itself may not be able to regulate AKT signaling in a similar manner to low K. While we are concluding that our data overall support a role for CI in regulating mTOR/AKT signaling, we have adjusted our discussion to include a section on cell volume/molecular crowding as potentially playing a role that our current data cannot fully exclude. We have additionally mentioned that we are currently unable to measure PT cell CI as a limitation of our study. We are working to establish models for this purpose similar to approaches that have already been used for distal nephron segments.¹

2) Fig 1&2. Overall, the first section could be more mechanistic. The results suggest that ENaC activation is driving the increase in blood Na in KO mice, and the results with amiloride treatment correcting blood Na⁺ are convincing (2F). But how? Does amiloride increase urine Na excretion without affecting urine volume (not typically)? Blood Na⁺ can also increase due to increases in free water clearance. What is the urine volumes and osms in these mice? How does it change with time and diet. Why do the KO mice have an increase in UNa (1I) while also having an increase in blood Na⁺ (Fig 1B)? For Fig 1 it would be useful to show both the concentration of Na⁺ and the total Na⁺ based on urine volume. A limitation for the spot urines is that total Na⁺ cannot be determined. For Fig 2 what was UNa/Cr with amiloride treatment.

We have now calculated free water clearance in control and KO animals and included these data as figure 1L. These data confirm that KO mice retain less water relative to controls and this underlies the increased blood Na. We have also included the urine K and Na concentrations, as requested, in our new figs 1e and i.

For figure 2, we have also included the UNa/Cr with amiloride in knockout mice (new supplemental figure 3a) and have shown that while amiloride did reduce urine K in knockout mice, it did not have a detectable effect on urine Na. We hypothesize that ENaC activity in knockout animals, while likely minimal on low K, is enough to drive K wasting on a 0K diet, but is not a major contributor to Na wasting because other distal nephron transport pathways are more dominant.

We have also added UOsm data from our amiloride-treated and untreated KO mice on 0K diet (new supplemental fig 3b). Higher UOsm with amiloride supports the observed effect on blood Na with ENaC blockade. The mechanisms underlying how ENaC inhibition causes this in our knockout animals and the underlying proximal-distal nephron signaling, while not the primary focus of this manuscript, is a current focus of our lab and will be explored in subsequent work.

3) Fig 3: It's not clear to me what is driving the increase in ENaC, pNCC, and tNKCC2 activity. If the Kir4.2 KO mice have a trend towards lower serum K⁺ on control diet that could increase pNCC, but not ENaC and tNKCC2. This seems short on mechanism.

We have now included data that Kir4.2 KO mice have elevated aldosterone (new figure 3c) and retain less water (new fig 1L) relative to control animals. These metabolic and hormonal changes provide an explanation for the ENaC and NKCC2 activation. This has also been addressed in our discussion .

4) Fig 5C: Which AKT antibody was used for this? Is the punctate staining similar for AKT1 and AKT2? What does AKT staining look like in the Kir 4.2 KO mouse on control and low K⁺? If the signal is similar, what does it mean?

This was a pan AKT antibody that recognizes all AKT isoforms. The punctate staining for AKT1 and AKT2 is similar and we have added this to our data supplement (supplemental fig 7). The

lack of a change in AKT localization upon low K stimulation or Kir4.2 knockout indicates that altered AKT localization is not a key signal underlying our mechanism. We have added this point to our discussion.

5) Fig 5 D & E, Sup Fig 3G & H: Are these figures showing the same data from the same mice? If so then what do the values differ for for urine K⁺ in WT mice fed OK. The MK-2206 mice in Fig 5 have a max value ~2umol, whereas the MK-2206 mice in Sup Fig 3 have a max value of ~4.5umol.

Figure 5D and E depict the MK-2206-induced natriuretic and kaliuretic responses in wild-type animals on normal K and a OK diets. Supplemental 3G and H in the initial version (Supplemental 8B and 8C in revised submission) show the same responses, but are comparing the OK responses to the NK response. In these panels we are interested in determining the fold-change relative to vehicle so the vehicle data have been normalized to a value of 1. This enables a clear comparison of the relative increases (vs vehicle) on a NK or OK diet. This has been clarified in the figure legend.

6) In the first section the authors state that distal transport pathways compensate for proximal Na⁺ losses in Kir4.2 KO mice. In the second section the authors state that urine Na⁺ losses with MK-2206 treatment are not dependent on the distal pathway based on Fig 5G. However, this isn't clear why the distal pathway would be involved in the first instance and not the second.

In the chronic setting of Kir4.2 deletion, we do hypothesize that distal pathways are compensatory for a reduced proximal tubule transport capacity. Here we are highlighting that the kaliuretic and natriuretic responses to AKT inhibition acutely are due to proximal inhibition of electrolyte transport rather than purely a distal phenomenon. A key difference between the two conditions is that the diuresis studies are acute, which likely does not allow enough time for a distal compensatory response. If there were a major CNT effect of the compound on principal cell AKT, we would expect an increase in the Urine Na/K ratio due to ENaC inhibition. This was not observed and therefore are concluding that the principal cell is not the major site of action of MK2206 in our model. This has been added to our discussion section for clarification.

7) Based on the NIH new Data Management and Sharing policy, please include details re: where the results from the phosphoproteomics study will be shared.

This information has been added under the 'Data Availability section.'

Minor:

1) line 320: "Large differences in blood electrolytes were not detected...consumptions of Na⁺ deficient diet". Yet, significant differences were seen in blood [Cl⁻] so this statement is not accurate. Please clarify.

The text has now been adjusted to reflect our observance of blood Cl changes.

2) line 330: "While 15-hour urine Na⁺ excretion was not different" Fig 1H. Is the 15h urine the amount of time it is collected? Please clarify what you mean by 15h since the X-axis is in days. Perhaps showing a treatment timeline would be helpful.

We have now detailed our collection protocol in the methods section. We have changed our nomenclature from “15-hour” urine to “daily urine” for clarification.

3) Fig 3: please indicate the diet these mice were on

It is now stated that these animals were maintained on a normal diet.

4) Fig 3K: please switch lasix to furosemide

We have made this change

5) Sup Fig 3E. Can you include a positive control for tAKT3 to show that the antibody works? Or cite Mark Kneppers paper PMID: 33769951, showing very low levels of AKT3 mRNA that supports your finding.

We have now done both

6) Fig 5C: please indicate on the figure/image the type of diet used

It is now stated in the figure and corresponding legend that these animals were maintained on a normal diet.

7) Fig Sup 3F: please include a higher magnification insert as the punctate cannot be appreciated in this version

This has now been included as supplemental fig 6g to highlight the punctate staining.

8) Fig Sup 3G: what is meant by "urine Na⁺ normalized to excretion", excretion of what?

Data in this panel and panel H (Supplemental fig 8B and 8C in revised submission) have been normalized to the vehicle-induced response to allow a clearer between-diet comparison of the MK-2206-induced responses. The average vehicle response has been set to a value of 1. Please see our response to major point #5 above for additional details. This has now been clarified in the figure legend.

9) Fig 5H: is this showing the concentration of Na⁺ per gBW at 4h? If so, why is there a negative value. Is it showing a change over 4h? Please clarify in the figure legend.

Each data point presented for H and I is the difference between MK-2206- and vehicle-treated electrolyte excretion for each animal. This has been added to the figure legend.

Reviewer #2 (Remarks to the Author)

This paper is an interesting extension of the authors previous description of the Kir4.2. knockout (previously published in Cell reports 2022). The authors aim to demonstrate that potassium homeostasis is altered. Unfortunately, the paper makes several claims that are not supported by

the mechanistic data presented. The mechanistic data largely relies on ex vivo tubules that are treated with very large and non-physiological concentrations of ions, and on an AKT inhibitor.

Comments:

- Background and sexual dimorphism, if any, of the mice is not reported.

This has now been included in the methods section.

- In fig. 2 they show that the excretion is dependent on ENaC, unfortunately only in short-term experiments and only using amiloride, an unspecific inhibitor.

We have now performed a similar experiment with a different ENaC inhibitor (K-sparing diuretic), triamterene, to demonstrate a similar reduction in K excretion in Kir4.2 KO mice. These data are consistent with our amiloride data and have been included as supplemental figure 3c. These data further support a role for ENaC activity as a mediator of the K wasting in Kir4.2 KO animals.

- In Fig. 3, the phosphorylation abundance (i.e. p-NCC) is regulated in the quantification, but not in the corresponding immunoblot. More careful immunoblot quantification is needed. Figure 3: What age did the mouse have?

We have repeated and included new blots for phosphorylated and total NCC in figure 3. Mice were all aged 8-12 weeks for all experiments. We have included this in the methods section.

- In Fig. 4, the LTL projections are not very useful or very clear, since the LTL seems to label different parts of the cell – would this be also visible in normal EM/proximal tubule histology?

We have repeated our LTL staining along with the Na/K ATPase to clearly delineate apical and basolateral membranes in the PT of our mice. We have performed quantification of PT tubule diameter using these new images and have added these data to our data supplement (Supplemental fig 5). These new data support our previous findings in figure 4 that Kir4.2 KO mice have reduced PT tubule diameters.

- Figure 5C: The IF is of low quality. Please improve staining quality.

This has been improved and we have added a higher zoom panel as new figure 5d

- Understanding blood pressure and activation of the renin-angiotensin-aldosterone systems, particular renin and aldosterone, are important to understand and support the mechanisms postulated here. Unfortunately, the blood pressure measurements are of low quality, because training was only done for 3 days (Suppl Fig. 1), and tail cuffs instead of telemetry were used.

We have performed catheter-based invasive blood pressure measurements and included these data in supplemental figure 1i. Aldosterone has also been measured and included as figure 3c.

- Phosphoproteomics results cannot be interpreted without proteomics data. I.e. a phosphorylation switch might be real, or just a result of altered protein abundance. Data on proteome level, at least for representative phosphorylation sites important for nutrient sensing, should be provided. In addition using top 10 percentile is really not appropriate. One should use statistics with corrected false-discovery rates to make sure that these regulations are real. Similarly, the analysis was done using IPA analysis on protein level – not on the site level. It is not known if any of the regulated sites are functional, and bona-fide enriched kinase subsets for Akt/mTOR.

We have now included the total proteomics dataset and the statistical analyses as part of our data that has been deposited into the ProteomeXchange database and included as Supplemental table 1 respectively. We have also clarified in our text that we used the less stringent 10th percentile as a hypothesis generating initiative to not exclude potential candidate pathways. This approach was agreed upon by our team to make this cost feasible and also minimize loss of animal life at a hypothesis-generating stage of the study. While phosphospecific antibodies were not available for all identified sites, we were able to use the phosphoproteomics data to drive a pathway-confirming line of inquiry in figures 7-9. We have made this clear in the text and highlighted that confirmatory studies on each of the novel identified sites will be pursued in future study. Regarding IPA analysis, it was performed at the phosphorylation site level.

- AKT function: The authors are not able to prove involvement of Akt in vivo. A high dose of the M2260 inhibitor (probably 100x of the human dose) is used, likely causing off-target effects. The AKT inhibitor M2260 would expect to alter blood glucose, which might change proximal tubule function, or diarrhea which would alter electrolyte balance. <https://ascopubs.org/doi/10.1200/JCO.2011.35.5263> Additional inducible genetic models are likely necessary to show involvement of AKT.

We agree that off-target effects cannot be ruled out by our analysis. However, we chose our dose to be well below those used in prior animal reports.^{2,3} We have performed additional experiments at a lower dose of the inhibitor, which yielded results consistent with our original experiment. These new data have been included as new supplemental figure 8a.

We agree with the reviewer on the essential nature of inducible genetic models for future study of epithelial AKT function in the kidney. Our group is currently performing these studies for future reports that are beyond the scope of this manuscript.

It should also be noted that cross-species dosing conversion must consider multiple factors beyond a mere comparison of dose-to-body mass ratio, including body surface area-to-weight ratio as well as species-specific metabolic rates, both of which are much greater in mice than in humans.⁴ The 100-fold higher dose estimate mentioned by the reviewer is likely an overestimate.

- Figure 7: mTOR activity is also regulated by phosphorylation, but particularly also by interaction as mTORC1 and mTORC2. This information on mTOR is missing, and would likely have to be tested by additional genetic studies. Like this, the altered phosphorylation remains an association, not a proof.

We agree with the reviewer and have now clearly noted the complexity of the mTORC1 and mTORC2 complexes in mTOR regulation in our discussion. We have highlighted one component of the regulation in this manuscript, but our group is currently performing in vivo studies of the two complexes for future reports.

- Figure 8: Virtually all pH and ion concentrations are not in physiological range and thus, the regulations shown here are likely not physiologically relevant.

We have now included a potassium titration curve as figure 8b to show the effects of extracellular K are graded throughout the physiological range. For all other variables, including pH, HCO₃, and Cl, we have included groups both within and beyond the physiological range. The more extreme treatments are essential to understand the biology and how the system responds when pushed to limits. A lack of responsiveness to both pH and HCO₃, for example, at extreme values tell us these variables are not highly influential on mTOR/AKT activation. We have performed regression analyses to confirm observations within the physiological range and beyond for thoroughness.

Reviewer #3 (Remarks to the Author):

The maintaining of systemic K⁺ levels is strongly associated with renal hypertrophy. Therefore, the identification of mediators and characterization of mechanisms behind hypokalemia-linked hypertrophy is crucial in preventing salt-sensitive hypertension. Previous studies established the involvement of distal tubule in salt-sensitive hypertension via distal activation of Na⁺ transport in response to inwardly rectifying K⁺ channel, Kir4.1 mediated low K⁺ states. In this elegant study, the authors identified Kir4.2 as a mediator of the proximal tubule response to K⁺ deficiency. The authors observed that removal of K⁺ from diet developed proximal renal tubular acidosis in knockout animals and increased urinary K⁺ excretion caused by ENaC-mediated increased distal sodium delivery. Kir4.2 has a role in proximal tubule expansion through mTORC2/AKT signaling. Moreover, they recognized intracellular Cl⁻ as a second messenger activating mTORC2/AKT mTORC2/AKT to coordinate cell growth and electrolyte transport in the low K⁺ response.

This study is comprehensive and exciting, and the findings are novel. The study should have significant value in understanding Kir4.2-mediated intracellular proximal ions homeostasis in regulating salt-sensitive hypertension. The experiments are well-designed and described. I have several comments, as listed below:

1. As Kir4.1 mediates electrogenic basolateral K⁺ efflux to coordinate extracellular K⁺ changes, what is the status of Kir4.1 in Kir4.2^{-/-} mice? Does the abundance of Kir4.1 increase when Kir4.2 is deleted?

We have included these new data as new supplemental fig 3. Kir4.1 abundance did not differ between genotypes.

2. Why does Na⁺ deficient diet have lower Cl⁻ (0.07%) than K⁺-deficient (0.45% Cl⁻)?

This is because Na has been removed in the form of NaCl for the Na-deficient diet. The text has been adjusted for accuracy to indicate we are using a NaCl-deficient diet.

3. 'Renal mTORC2/AKT signaling is activated via changes in cytosolic Cl⁻ in response to reduced extracellular K⁺.' Is the Cl⁻ compartmentalized or cytosolic? It is more convenient to state intracellular Cl⁻ instead of cytosolic Cl⁻. What is the critical potassium level in the kidney that could cause Cl⁻-induced mTOR activation? What if the intracellular K⁺ and Cl⁻ levels in Kir4.2^{+/+} and Kir4.2^{-/-} mice were on a normal diet and 0K diet?

We agree with the reviewer's point here and have changed cytosolic to intracellular where appropriate. According to our ex vivo tubule data included as Figure 8b, there is no threshold or critical potassium level that causes low Cl⁻-induced mTOR activation. Rather, this effect is graded throughout the physiological range.

Unfortunately, we are unable to measure intracellular K and Cl levels in animals at this time. There is evidence that transgenic mice expressing Cl⁻-sensitive fluorescent probes can be used as a tool to measure intracellular Cl in the kidney as this approach has been attempted to study the distal nephron.¹ Our group is currently investigating this approach for use along the proximal tubule.

4. Fig. 3: Phospho and total NCC bands are not clearly visible, while others found prominent bands for these proteins (PMID: 36719746). Replace these blots with more visible bands.

We have repeated and included new blots for phosphorylated and total NCC in figure 3.

5. Fig. 7C: The pNDRG1 band is not clear. Replace the blot.

This has been repeated and replaced in figure 7.

Minor comments:

1. Supplemental Figure 5: Western blot quantification for figure 8b-d. But the quantifications are for Figure 7b-d.

This has been changed.

2. Fig4A: 'Kidney mass from Kir4.2^{+/+} and Kir4.2^{-/-} mice on a normal diet and 0K diets at indicated timepoints.' However, the data showed only one plot for WT and KO. According to the result section description, it seems to be a 0K diet. Need clarification.

Figure 4a shows 3 timepoints, 0 days, 4 days, and 8 days of 0K feeding (with 0 days being equivalent to a normal diet) in Kir4.2^{+/+} and Kir4.2^{-/-} animals. Each data point in the graph is the average of n animals indicated for that timepoint in the figure legend.

1. Su XT, Klett NJ, Sharma A, et al. Distal convoluted tubule Cl(-) concentration is modulated via K(+) channels and transporters. *Am J Physiol Renal Physiol* 2020;319(3):F534-F540. DOI: 10.1152/ajprenal.00284.2020.
2. Hirai H, Sootome H, Nakatsuru Y, et al. MK-2206, an allosteric Akt inhibitor, enhances antitumor efficacy by standard chemotherapeutic agents or molecular targeted drugs in vitro and in vivo. *Mol Cancer Ther* 2010;9(7):1956-67. DOI: 10.1158/1535-7163.MCT-09-1012.
3. Winder A, Unno K, Yu Y, Lurain J, Kim JJ. The allosteric AKT inhibitor, MK2206, decreases tumor growth and invasion in patient derived xenografts of endometrial cancer. *Cancer Biol Ther* 2017;18(12):958-964. DOI: 10.1080/15384047.2017.1281496.
4. Nair AB, Jacob S. A simple practice guide for dose conversion between animals and human. *J Basic Clin Pharm* 2016;7(2):27-31. DOI: 10.4103/0976-0105.177703.

REVIEWER COMMENTS

Reviewer #1 (Remarks to the Author):

Overall, Zhang et al thoughtfully addressed the concerns that were identified. The first half of the paper flows better in to the second half and the observations support their hypothesis. This continues to be an interesting and important manuscript, but some concerns remain.

1) It is appreciated that in the discussion, the authors mention the limitation of not being able to measure Cl⁻ in the proximal tubule. This is a problem relative to the claims in the manuscript, as it remains unclear whether low K⁺ changes proximal tubule intracellular [Cl⁻] to any major degree. Thus, the importance of intracellular chloride concentration in this process remains overstated. As mentioned in the previous review, in the absence of chloride measurements, or any clear mechanism supporting chloride “second messenger” inhibition of mTOR, these claims need to be modified. The helpful hypertonicity experiments argue against a short-term effect of crowding and support a short term effect of reduced ionic strength, but as mentioned by the authors, they are insufficient to rule out non-chloride mechanisms, or crowding-dependent mechanisms that occur across longer timescales, such as mTOR dependent mechanisms that mediate hypertrophy. With this in mind, the following amendments to the manuscript are recommended:

- In the abstract: “...secondary changes in intracellular Cl⁻“ should be changed to “secondary changes in Cl⁻ transport”, as the former implies a change in intracellular chloride concentration that was not measured.
- The statement in the last paragraph of the introduction should be modified as it is not supported by evidence: “they support cell Cl⁻ as a second messenger activating mTORC2/AKT”.
- In the Discussion section, “Intracellular Cl⁻ is a K⁺ responsive ion...”, there needs to be a consideration of how the short-term isolated tubule suspension measurements over 30 minutes relate to long-term dietary maneuvers over days. It’s quite possible that they don’t.
- The model in Figure 10 needs to be modified. Specifically, since intracellular chloride was not measured, please remove the “decreased [Cl⁻]_i” from the diagram, and draw an arrow from the voltmeter to mTORC2 with an accompanying question mark.

Other minor comments:

2) In either the results or Fig 2C please include the amiloride dose used

3) In the results section discussing figure 5c, d please include "In Kir4.2+/+ mice"...AKT staining adopted a punctate pattern in the PT both at baseline and following OK feeding.

4) In either the results or Fig 5e please include the duration of MK2206 treatment.

5) What was the effect of MK2206 on urine volume in the Kir4.2-/- mice on OK diet (connected to Fig 5i and Fig 5j)?

6) Please include citation in the last paragraph of the results for "The WNK kinases, while being modulated by Cl⁻, are also known to be molecular crowding sensors and are activated by cell shrinkage".

7) Fig 4C, the labels for Kir4.2 in the cleared kidneys are difficult to see

8) Figure 8I, the label "tubule suspensions" is mislocated

Reviewer #2 (Remarks to the Author):

The authors have addressed my concerns except for one: Some statistical testing must be applied to phosphoproteomics data, and not a "10% percentile cutoff". This reviewer feels that "costs" or "loss of animal life" are very noble motivations, but do not warrant large scale data without statistics (p-value/FDR) and low numbers of replicates (n=2/3). In this case, I do not think these data are valid and suggest to remove the data from the paper.

Reviewer #3 (Remarks to the Author):

The authors revised the manuscript thoroughly and adequately addressed all my criticisms. I have no further concerns. The study is comprehensive, meritorious, and suitable for the nature communication audience.

REVIEWER COMMENTS

Reviewer #1 (Remarks to the Author):

Overall, Zhang et al thoughtfully addressed the concerns that were identified. The first half of the paper flows better in to the second half and the observations support their hypothesis. This continues to be an interesting and important manuscript, but some concerns remain.

1) It is appreciated that in the discussion, the authors mention the limitation of not being able to measure Cl⁻ in the proximal tubule. This is a problem relative to the claims in the manuscript, as it remains unclear whether low K⁺ changes proximal tubule intracellular [Cl⁻] to any major degree. Thus, the importance of intracellular chloride concentration in this process remains overstated. As mentioned in the previous review, in the absence of chloride measurements, or any clear mechanism supporting chloride “second messenger” inhibition of mTOR, these claims need to be modified. The helpful hypertonicity experiments argue against a short-term effect of crowding and support a short term effect of reduced ionic strength, but as mentioned by the authors, they are insufficient to rule out non-chloride mechanisms, or crowding-dependent mechanisms that occur across longer timescales, such as mTOR dependent mechanisms that mediate hypertrophy. With this in mind, the following amendments to the manuscript are recommended:

- In the abstract: “...secondary changes in intracellular Cl⁻” should be changed to “secondary changes in Cl⁻ transport”, as the former implies a change in intracellular chloride concentration that was not measured.

This change has been made.

- The statement in the last paragraph of the introduction should be modified as it is not supported by evidence: “they support cell Cl⁻ as a second messenger activating mTORC2/AKT”.

This has been toned down and changed in accordance with the above comment to mention Cl transport and not Cl concentration.

- In the Discussion section, “Intracellular Cl⁻ is a K⁺ responsive ion...”, there needs to be a consideration of how the short-term isolated tubule suspension measurements over 30 minutes relate to long-term dietary maneuvers over days. It’s quite possible that they don’t.

We have now added this limitation to our discussion and stated that we will need to determine how our acute studies relate to longer dietary models.

- The model in Figure 10 needs to be modified. Specifically, since intracellular chloride was not measured, please remove the “decreased [Cl⁻]_i” from the diagram, and draw an arrow from the voltmeter to mTORC2 with an accompanying question mark.

This change has been made

Other minor comments:

2) In either the results or Fig 2C please include the amiloride dose used

This has now been included in the results section.

3) In the results section discussing figure 5c, d please include "In Kir4.2+/+ mice"...AKT staining adopted a punctate pattern in the PT both at baseline and following OK feeding.

This has now been added.

4) In either the results or Fig 5e please include the duration of MK2206 treatment.

We have now included this in the results section.

5) What was the effect of MK2206 on urine volume in the Kir4.2-/- mice on OK diet (connected to Fig 5i and Fig 5j)?

It tended to be lower in the knockouts, but did not achieve statistical significance. We have now included these data as Supplemental figure 8d.

6) Please include citation in the last paragraph of the results for "The WNK kinases, while being modulated by Cl⁻, are also known to be molecular crowding sensors and are activated by cell shrinkage".

This has now been added

7) Fig 4C, the labels for Kir4.2 in the cleared kidneys are difficult to see

we have now increased the text size in Fig 4b and c.

8) Figure 8l, the label "tubule suspensions" is mislocated

This has been corrected

Reviewer #2 (Remarks to the Author):

The authors have addressed my concerns except for one: Some statistical testing must be applied to phosphoproteomics data, and not a "10% percentile cutoff". This reviewer feels that "costs" or "loss of animal life" are very noble motivations, but do not warrant large scale data without statistics (p-value/FDR) and low numbers of replicates (n=2/3). In this case, I do not think these data are valid and suggest to remove the data from the paper.

These data have now been removed from the paper.

Reviewer #3 (Remarks to the Author):

The authors revised the manuscript thoroughly and adequately addressed all my criticisms. I have no

further concerns. The study is comprehensive, meritorious, and suitable for the nature communication audience.

REVIEWERS' COMMENTS

Reviewer #1 (Remarks to the Author):

The authors have addressed all of our remaining concerns with this latest revision. Our congratulations on the completion of a well-executed, thought-provoking, and important study.

Ro Subramanya & Cary Boyd-Shiwarski